# Large Stepsize Gradient Descent for Non-Homogeneous Two-Layer Networks: Margin Improvement and Fast Optimization

**Yuhang Cai**[1]    **Jingfeng Wu**[1]    **Song Mei**[1]    **Michael Lindsey**[1,2]    **Peter L. Bartlett**[1,3]

[1]UC Berkeley    [2]Lawrence Berkeley National Laboratory    [3]Google DeepMind

{willcai,uuujf,songmei,lindsey,peter}@berkeley.edu

## Abstract

The typical training of neural networks using large stepsize gradient descent (GD) under the logistic loss often involves two distinct phases, where the empirical risk oscillates in the first phase but decreases monotonically in the second phase. We investigate this phenomenon in two-layer networks that satisfy a near-homogeneity condition. We show that the second phase begins once the empirical risk falls below a certain threshold, dependent on the stepsize. Additionally, we show that the normalized margin grows nearly monotonically in the second phase, demonstrating an implicit bias of GD in training non-homogeneous predictors. If the dataset is linearly separable and the derivative of the activation function is bounded away from zero, we show that the average empirical risk decreases, implying that the first phase must stop in finite steps. Finally, we demonstrate that by choosing a suitably large stepsize, GD that undergoes this phase transition is more efficient than GD that monotonically decreases the risk. Our analysis applies to networks of any width, beyond the well-known neural tangent kernel and mean-field regimes.

## 1 Introduction

Neural networks are mostly optimized by *gradient descent* (GD) or its variants. Understanding the behavior of GD is one of the key challenges in deep learning theory. However, there is a nonnegligible discrepancy between the GD setups in theory and in practice. In theory, GD is mostly analyzed with relatively small stepsizes such that its dynamics are close to the continuous *gradient flow* dynamics, although a few exceptions will be discussed later. However, in practice, GD is often used with a relatively large stepsize, with behaviors significantly deviating from that of small stepsize GD or gradient flow. Specifically, notice that small stepsize GD (hence also gradient flow) induces monotonically decreasing empirical risk, but in practice, good optimization and generalization performance is usually achieved when the stepsize is large and the empirical risk oscillates [see Wu and Ma, 2018, Cohen et al., 2020, for example]. Therefore, it is unclear which of the theoretical insights drawn from analyzing small stepsize GD apply to large stepsize GD used practically.

The behavior of small stepsize GD is relatively well understood. For instance, classical optimization theory suggests that GD minimizes convex and $L$-smooth functions if the stepsize $\tilde{\eta}$ is well below $2/L$, with a convergence rate of $\mathcal{O}(1/(\tilde{\eta}t))$, where $t$ is the number of steps [Nesterov, 2018]. More recently, Soudry et al. [2018], Ji and Telgarsky [2018] show an *implicit bias* of small stepsize GD in logistic regression with separable data, where the direction of the GD iterates converges to the max-margin direction. Subsequent works extend their implicit bias theory from linear model to *homogenous* networks [Lyu and Li, 2020, Chizat and Bach, 2020, Ji and Telgarsky, 2020]. These theoretical results all assume the stepsize of GD is small (and even infinitesimal) such that the empirical risk decreases monotonically and, therefore cannot be directly applied to large stepsize GD used in practice.

38th Conference on Neural Information Processing Systems (NeurIPS 2024).

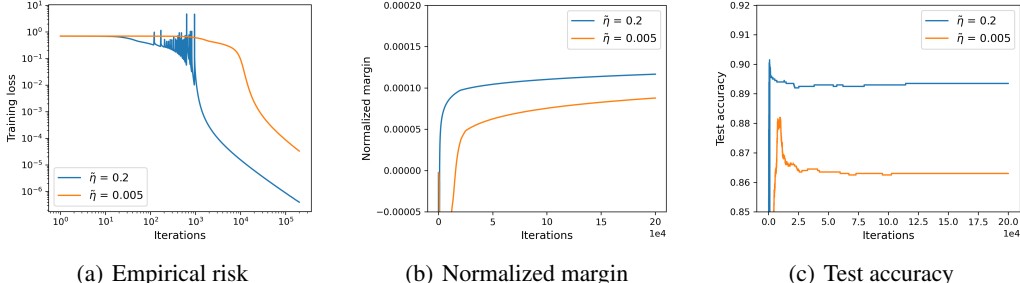

(a) Empirical risk       (b) Normalized margin       (c) Test accuracy

Figure 1: The behavior of (GD) for optimizing a non-homogenous four-layer MLP with GELU activation function on a subset of CIFAR-10 dataset. We randomly sample $6,000$ data with labels "airplane" and "automobile" from CIFAR-10 dataset. The normalized margin is defined as $\min_{i\in[n]} y_i f(\mathbf{w}_t; \mathbf{x}_i)/\|\mathbf{w}_t\|^4$, which is close to (3). The blue curves correspond to GD with a large stepsize $\tilde{\eta} = 0.2$, where the empirical risk oscillates in the first phase but decreases monotonically in the second phase. The orange curves correspond to GD with a small stepsize $\tilde{\eta} = 0.005$, where the empirical risk decreases monotonically. Furthermore, Figure 1(b) suggests the normalized margins of both two curves increase and converge in the stable phases. Finally, Figure 1(c) suggests that large stepsize achieves a better test accuracy, consistent with larger-scale learning experiment [Hoffer et al., 2017, Goyal et al., 2017]. More details can be found in Section 5.

More recently, large stepsize GD that induces oscillatory risk has been analyzed in simplified setups [see Ahn et al., 2023, Zhu et al., 2022, Kreisler et al., 2023, Chen and Bruna, 2023, Wang et al., 2022a, Wu et al., 2023, 2024, for an incomplete list of references]. In particular, in logistic regression with linearly separable data, Wu et al. [2023] showed that the implicit bias of GD (that maximizes the margin) holds not only for small stepsizes [Soudry et al., 2018, Ji and Telgarsky, 2018] but also for an arbitrarily large stepsize. In the same problem, Wu et al. [2024] further showed that large stepsize GD that undergoes risk oscillation can achieve an $\tilde{\mathcal{O}}(1/t^2)$ empirical risk, whereas small stepsize GD that monotonically decreases the empirical risk must suffer from a $\Omega(1/t)$ empirical risk. Nonetheless, these theories of large stepsize GD are limited to relatively simple setups such as linear models. The theory of large stepsize GD for nonlinear networks is underdeveloped.

This work fills the gap by providing an analysis of large stepsize GD for nonlinear networks. In the following, we set up our problem formally and summarize our contributions.

**Setup.** Consider a binary classification dataset $(\mathbf{x}_i, y_i)_{i=1}^n$, where $\mathbf{x}_i \in \mathbb{R}^d$ is a feature vector and $y_i \in \{\pm 1\}$ is a binary label. For simplicity, we assume $\|\mathbf{x}_i\| \leq 1$ for all $i$ throughout the paper. For a predictor $f$, the empirical risk under logistic loss is defined as

$$L(\mathbf{w}) := \frac{1}{n}\sum_{i=1}^{n} \ell(y_i f(\mathbf{w}; \mathbf{x}_i)), \quad \ell(t) := \log(1 + e^{-t}). \tag{1}$$

Here, the predictor $f(\mathbf{w}; \cdot) : \mathbb{R}^d \mapsto \mathbb{R}$ is parameterized by trainable parameters $\mathbf{w}$ and is assumed to be continuously differentiable with respect to $\mathbf{w}$. The predictor is initialized from $\mathbf{w}_0$ and then trained by *gradient descent* (GD) with a constant stepsize $\tilde{\eta} > 0$, that is,

$$\mathbf{w}_{t+1} := \mathbf{w}_t - \tilde{\eta}\nabla L(\mathbf{w}_t), \quad t \geq 0. \tag{GD}$$

We are interested in a nonlinear predictor $f$ and a large stepsize $\tilde{\eta}$. A notable example in our theory is two-layer networks with Lipschitz, smooth, and nearly homogenous activations (see (2)). Note that minimizing $L(\mathbf{w})$ is a non-convex problem in general.

**Observation.** Empirically, large stepsize GD often undergoes a phase transition, where the empirical risk defined in (1) oscillates in the first phase but decreases monotonically in the second phase (see empirical evidence in Appendix A in [Cohen et al., 2020] and a formal proof in [Wu et al., 2024] for linear predictors). This is illustrated in Figure 1. We follow Wu et al. [2024] and call the two phases the *edge of stability* (EoS) phase [name coined by Cohen et al., 2020] and the *stable* phase, respectively.

**Contributions.** We prove the following results for large stepsize GD for training nonlinear predictors under logistic loss.

1. For Lipschitz and smooth predictor $f$ trained by GD with stepsize $\tilde{\eta}$, we show that as long as the empirical risk is below a threshold depending on $\tilde{\eta}$, GD monotonically decreases the empirical risk (see Theorem 2.2). This result extends the stable phase result in Wu et al. [2024] from linear predictors to nonlinear predictors, demonstrating the generality of the existence of a stable phase.

2. Assuming that GD enters the stable phase, if in addition the preditor has a bounded homogenous error (see Assumption 1C), we show that the normalized margin induced by GD, $\min_i y_i f(\mathbf{w}_t; \mathbf{x}_i)/\|\mathbf{w}_t\|$, nearly monotonically increases (see Theorem 2.2). To the best of our knowledge, this is the first characterization of implicit bias of GD for *non-homogenous* predictors. In particular, our theory covers two-layer networks with commonly used activations functions (which are often non-homogenous) that cannot be covered by existing results [Lyu and Li, 2020, Ji and Telgarsky, 2020, Chizat and Bach, 2020].

3. Under additional technical assumptions (the dataset is linearly separable and the derivative of the activation function is bounded away from zero), we show that the initial EoS phase must stop in $\mathcal{O}(\tilde{\eta})$ steps and GD transits to the stable phase afterwards. Furthermore, by choosing a suitably large stepsize, GD achieves a $\tilde{\mathcal{O}}(1/t^2)$ empirical risk after $t$ steps. In comparison, GD that converges monotonically incurs an $\Omega(1/t)$ risk. This result indicates an optimization benefit of using large stepsize and generalizes the results in [Wu et al., 2024] from linear predictors to neural networks.

## 2 Stable Phase and Margin Improvement

In this section, we present our results for the stable phase of large stepsize GD in training nonlinear predictors. Specifically, our results apply to nonlinear predictors that are Lipschitz, smooth, and nearly homogeneous, as described by the following assumption.

**Assumption 1** (Model conditions). *Consider a predictor $f(\mathbf{w}; \mathbf{x}_i)$, where $\mathbf{x}_i$ is one of the feature vectors in the training set.*

A. **Lipschitzness**. *Assume there exists $\rho > 0$ such that for every $\mathbf{w}$, $\sup_i \|\nabla_\mathbf{w} f(\mathbf{w}; \mathbf{x}_i)\| \leq \rho$.*

B. **Smoothness**. *Assume there exists $\beta > 0$ such that for all $\mathbf{w}, \mathbf{v}$,*

$$\|\nabla f(\mathbf{w}; \mathbf{x}_i) - \nabla f(\mathbf{v}; \mathbf{x}_i)\| \leq \beta\|\mathbf{w} - \mathbf{v}\|, \quad i = 1, \ldots, n.$$

C. **Near homogeneity**. *Assume there exists $\kappa > 0$ such that for every $\mathbf{w}$,*

$$|f(\mathbf{w}; \mathbf{x}_i) - \langle \nabla_\mathbf{w} f(\mathbf{w}; \mathbf{x}_i), \mathbf{w} \rangle| \leq \kappa, \quad i = 1, \ldots, n.$$

Assumptions 1A and 1B are commonly used conditions in the optimization literature. Note that Assumption 1B implies continuous differentiability, thus ruling out networks with ReLU activation function. The continuous differentiability is only used in our current stable phase analysis. We conjecture it can be relaxed using subdifferentiability [Lyu and Li, 2020] for allowing ReLU networks.

If $\kappa = 0$, then Assumption 1C requires the predictor to be exactly 1-homogenous. Our Assumption 1C allows the predictor to have a bounded homogenous error. It is clear that linear predictors $f(\mathbf{w}; \mathbf{x}) := \mathbf{w}^\top \mathbf{x}$ satisfy Assumption 1 with $\rho = \sup_i \|\mathbf{x}_i\| \leq 1$, $\beta = 0$, and $\kappa = 0$. Another notable example is *two-layer networks* given by

$$f(\mathbf{w}; \mathbf{x}) := \frac{1}{m} \sum_{j=1}^m a_j \phi(\mathbf{x}^\top \mathbf{w}^{(j)}), \quad \mathbf{w}^{(j)} \in \mathbb{R}^d, \quad j = 1, \ldots, m, \tag{2}$$

where we assume $a_j \in \{\pm 1\}$ are fixed and $\mathbf{w} \in \mathbb{R}^{md}$, the stack of $(\mathbf{w}^{(j)})_{j=1}^m$, are the trainable parameters. We define two-layer networks with the mean-field scaling [Song et al., 2018, Chizat and Bach, 2020, Chen et al., 2022, Suzuki et al., 2023]. However, our results hold for any width. The effect of rescaling the model will be discussed in Section 4. The following example shows that Assumption 1 covers two-layer networks with many commonly used activations $\phi(\cdot)$. The proof is provided in Appendix F.1.

**Example 2.1** (Two-layer networks). *Two-layer networks defined in* (2) *with the following activation functions satisfy Assumption 1 with the described constants:*

- **GELU**. $\phi(x) := \frac{x}{2}\mathrm{erf}\left(1 + (x/\sqrt{2})\right)$ *with* $\kappa = e^{-1/2}/\sqrt{2\pi}$, $\beta = 2/m$, *and* $\rho = (\sqrt{2\pi} + e^{-1/2})/\sqrt{2\pi m}$.

- **Softplus**. $\phi(x) := \log(1 + e^x)$ *with* $\kappa = \log 2$, $\beta = 1/m$, *and* $\rho = 1/\sqrt{m}$.

- **SiLU**. $\phi(x) := x/(1 + e^{-x})$ *with* $\kappa = 1$, $\beta = 4/m$, *and* $\rho = 2/\sqrt{m}$.

- **Huberized ReLU** *[Chatterji et al., 2021]. For a fixed* $h > 0$,

$$\phi(x) := \begin{cases} 0 & x < 0, \\ \frac{x^2}{2h} & 0 \leq x \leq h, \\ x - \frac{h}{2} & x > h, \end{cases}$$

*with* $\kappa = h/2$, $\beta = 1/(hm)$, *and* $\rho = 1/\sqrt{m}$.

**Margin for nearly homogenous predictors.** For a nearly homogenous predictor $f(\mathbf{w}; \cdot)$ (see Assumption 1C), we define its *normalized margin* (or *margin* for simplicity) as

$$\bar{\gamma}(\mathbf{w}) := \frac{\min_{i \in [n]} y_i f(\mathbf{w}; \mathbf{x}_i)}{\|\mathbf{w}\|}. \tag{3}$$

A large normalized margin $\bar{\gamma}(\mathbf{w})$ guarantees the prediction of each sample is away from the decision boundary. The normalized margin (3) is introduced by Lyu and Li [2020] for homogenous predictors. However, we show that the same notion is also well-defined for non-homogenous predictors that satisfy Assumption 1C. The next theorem gives sufficient conditions for large stepsize GD to enter the stable phase in training non-homogenous predictors and characterizes the increase of the normalized margin. The proof of Theorem 2.2 is deferred to Appendix A.

**Theorem 2.2** (Stable phase and margin improvement). *Consider* (GD) *with stepsize* $\tilde{\eta}$ *on a predictor* $f(\mathbf{w}; \mathbf{x})$ *that satisfies Assumptions 1A and 1B. If there exists* $r \geq 0$ *such that*

$$L(\mathbf{w}_r) \leq \frac{1}{\tilde{\eta}(2\rho^2 + \beta)}, \tag{4}$$

*then GD is in the stable phase for* $t \geq r$, *that is,* $(L(\mathbf{w}_t))_{t \geq r}$ *decreases monotonically. If additionally the predictor satisfies Assumption 1C and there exists* $s \geq 0$ *such that*

$$L(\mathbf{w}_s) \leq \min\left\{\frac{1}{e^{\kappa+2}2n}, \frac{1}{\tilde{\eta}(4\rho^2 + 2\beta)}\right\}, \tag{5}$$

*we have the following for* $t \geq s$:

- **Risk convergence**. $L(\mathbf{w}_t) = \Theta(1/t)$.

- **Parameter increase**. $\|\mathbf{w}_{t+1}\| \geq \|\mathbf{w}_t\|$ *and* $\|\mathbf{w}_t\| = \Theta(\log(t))$.

- **Margin improvement**. *There exists a modified margin function* $\gamma^c(\mathbf{w})$ *such that*

  - $\gamma^c(\mathbf{w}_t)$ *is increasing and bounded.*
  - $\gamma^c(\mathbf{w}_t)$ *is a multiplicative approximiator of* $\bar{\gamma}(\mathbf{w}_t)$, *that is, there exists* $c > 0$ *such that*

$$\gamma^c(\mathbf{w}_t) \leq \bar{\gamma}(\mathbf{w}_t) \leq \left(1 + \frac{c}{\log(1/L(\mathbf{w}_t))}\right)\gamma^c(\mathbf{w}_t), \quad t \geq s.$$

  *As a direct consequence,* $\lim_{t \to \infty} \bar{\gamma}(\mathbf{w}_t) = \lim_{t \to \infty} \gamma^c(\mathbf{w}_t)$.

Theorem 2.2 shows that for an arbitrarily large stepsize $\tilde{\eta}$, GD must enter the stable phase if the empirical risk falls below a threshold depending on $\tilde{\eta}$ given by (4). Furthermore, for nearly homogenous predictors, Theorem 2.2 shows that under a stronger risk threshold condition (5), the risk must converge at a $\Theta(1/t)$ rate and that the normalized margin nearly monotonically increases. This demonstrates an implicit bias of GD, even when used with a large stepsize and the trained predictor is non-homogenous.

**Limitations.** The stable phase conditions in Theorem 2.2 require GD to enter a sublevel set of the empirical risk. However, such a sublevel set might be empty. For instance, let $f(\mathbf{w}; \mathbf{x})$ be a two-layer network (2) with sigmoid activation. Notice that the predictor is uniformly bounded, $|f(\mathbf{w}; \mathbf{x})| \leq 1$, so we have

$$L(\mathbf{w}) = \frac{1}{n} \sum_{i=1}^{n} \log(1 + e^{-y_i f(\mathbf{w}; \mathbf{x}_i)}) \geq \log(1 + e^{-1}).$$

On the other hand, we can also verify that Assumption 1C is satisfied by $f(\mathbf{w}; \mathbf{x})$ with $\kappa = 1$ but no smaller $\kappa$. Therefore (5) cannot be satisfied. In general, the sublevel set given by the right-hand side of Assumption 1C is non-empty if

there exists a unit vector $\mathbf{v}$ such that $\min_i y_i f(\lambda \mathbf{v}; \mathbf{x}_i) \to \infty$ as $\lambda \to \infty$.

The above condition requires the data can be separated arbitrarily well by some predictor within the hypothesis class. This condition is general and covers (sufficiently large) two-layer networks (2) with many commonly used activations such as GeLU and SiLU. Moreover, although two-layer networks with sigmoid activation violate this condition, they can be modified by adding a leakage to the sigmoid to satisfy the condition. Furthermore, this condition can be satisfied for some nonlinear problems like XOR (or $k$-parity problems) since they can be realized by two-layer networks.

In the next section, we will provide sufficient conditions such that large stepsize GD will enter the stable phases characterized by (4) or (5).

**Comparisons to existing works.** Our Theorem 2.2 makes several important extensions compared to existing results [Wu et al., 2024, Lyu and Li, 2020, Chizat and Bach, 2020, Ji and Telgarsky, 2020]. First, Theorem 2.2 suggests that the stable phase happens for general nonlinear predictors such as two-layer networks, while the work by Wu et al. [2024] only studied the stable phase for linear predictors. Second, the margin improvement is only known for small (and even infinitesimal) stepsize GD and homogenous predictors [Lyu and Li, 2020, Chizat and Bach, 2020, Ji and Telgarsky, 2020], while we extend this to non-homogenous networks. To the best of our knowledge, Theorem 2.2 is the first implicit bias result covering large stepsize GD and non-homogenous predictors.

From a technical perspective, our proof uses techniques introduced by Lyu and Li [2020] for analyzing homogenous predictors. Our main innovation is the construction of new auxiliary margin functions that can deal with errors caused by large stepsize and non-homogeneity. More details are discussed in Appendix A.3.

## 3 Edge of Stability Phase

Our stable phase results in Theorem 2.2 require the risk to be below a certain threshold (see (4) and (5)). In this section, we show that the risk can indeed be below the required threshold, even when GD is used with large stepsize. Recall that minimizing the empirical risk with a nonlinear predictor is non-convex, therefore solving it by GD is hard in general. We make additional technical assumptions to conquer the challenges caused by non-convexity. We conjecture that these technical assumptions are not necessary and can be relaxed.

We focus on two-layer networks (2). We make the following assumptions on the activation function.

**Assumption 2** (Activation function conditions). *In the two-layer network* (2)*, let the activation function $\phi : \mathbb{R} \to \mathbb{R}$ be continuously differentiable. Moreover,*

A. **Derivative condition**. *Assume there exists $0 < \alpha < 1$ such that $\alpha \leq |\phi'(z)| \leq 1$.*

B. **Smoothness**. *Assume there exists $\tilde{\beta} > 0$ such that for all $x, y \in \mathbb{R}$, $|\phi'(x) - \phi'(y)| \leq \tilde{\beta}|x - y|$.*

C. **Near homogeneity**. *Assume there exists $\kappa > 0$ such that for every $z \in \mathbb{R}$, $|\phi(z) - \phi'(z)z| \leq \kappa$.*

Recall that $\sup_i \|\mathbf{x}_i\| \leq 1$. One can then check by direct computation that, under Assumption 2, two-layer networks (2) satisfy Assumption 1 with $\rho = 1/\sqrt{m}$, $\beta = \tilde{\beta}/m$, and $\kappa = \kappa$.

Assumptions 2B and 2C cover many commonly used activation functions. In Assumption 2A, we assume $|\phi'(z)| \leq 1$. This is just for the simplicity of presentation and our results can be easily

generalized to allow $|\phi'(z)| \leq C$ for a constant $C > 0$. The other condition in Assumption 2A, $|\phi'(z)| \geq \alpha$, however, is non-trivial. This condition is widely used in literature [see Brutzkus et al., 2018, Frei et al., 2021, and references thereafter] to facilitate GD analysis. Technically, this condition guarantees that each neuron in the two-layer network (2) will always receive a non-trivial gradient in the GD update; otherwise, neurons may be frozen during the GD update. Furthermore, commonly used activation functions can be combined with an identity map to satisfy Assumption 2A. This is formalized in the following example. The proof is provided in Appendix F.2.

**Example 3.1** (Leaky activation functions). *Fix* $0.5 \leq c < 1$.

- *Let* $\phi$ *be GELU, Softplus, or SilU in Example 2.1, then its modification* $\tilde{\phi}(x) := cx + (1-c)/4 \cdot \phi(x)$ *satisfies Assumption 2 with* $\kappa = 1$, $\alpha = 0.25$, *and* $\tilde{\beta} = 1$. *In particular, the modification of softplus can be viewed as a* smoothed *leaky ReLU.*

- *Let* $\phi$ *be the Huberized ReLU in Example 2.1, then its modification* $\tilde{\phi}(x) := cx + (1-c)/4 \cdot \phi(x)$ *satisfies Assumption 2 with* $\kappa = h/2$, $\alpha = 0.5$, *and* $\tilde{\beta} = 1/4h$.

- *The "leaky" tanh,* $\tilde{\phi}(x) := cx + (1-c)\tanh(x)$, *and the "leaky" sigmoid,* $\tilde{\phi}(x) := cx + c/(1+e^{-x})$, *both satisfy Assumption 2 with* $\kappa = 1, \alpha = 0.5$ *and* $\tilde{\beta} = 1$.

For the technical difficulty of non-convex optimization, we also need to assume a linearly separable dataset to conduct our EoS phase analysis.

**Assumption 3** (Linear separability). *Assume there is a margin* $\gamma > 0$ *and a unit vector* $\mathbf{w}_*$ *such that* $y_i \mathbf{x}_i^\top \mathbf{w}_* \geq \gamma$ *for every* $i = 1, \ldots, n$.

Assumption 3 serves as a sufficient condition for two-layer neural networks, regardless of width, to reach the initial bound of the stable phase under large stepsizes. We remark that our stable phase results do not need this assumption.

The following theorem shows that when GD is used with large stepsizes, the average risk must decrease even though the risk may oscillate locally.

**Theorem 3.2** (The EoS phase for two-layer networks). *Under Assumption 3, consider* (GD) *on two-layer networks* (2) *that satisfy Assumptions 2A and 2C. Denote the stepsize by* $\tilde{\eta} := m\eta$, *where* $m$ *is the network width and* $\eta$ *can be arbitrarily large. Then for every* $t > 0$, *we have*

$$\frac{1}{t}\sum_{k=0}^{t-1} L(\mathbf{w}_k) \leq \frac{1 + 8\log^2(\gamma^2 \eta t)/\alpha^2 + 8\kappa^2/\alpha^2 + \eta^2}{\gamma^2 \eta t} + \frac{\|\mathbf{w}_0\|^2}{m\eta t} = \mathcal{O}\left(\frac{\log^2(\eta t) + \eta^2}{\eta t}\right).$$

Theorem 3.2 suggests that the average risk of training two-layer networks decreases even when GD is used with large stepsize. Consequently, the risk thresholds (4) and (5) for GD to enter the stable phase must be satisfied after a finite number of steps. This will be discussed in depth in the next section.

Compared to Theorem 1 in [Wu et al., 2024], Theorem 3.2 extends their EoS phase bound from linear predictors to two-layer networks.

## 4 Phase Transition and Fast Optimization

For two-layer networks trained by large stepsize GD, Theorem 3.2 shows that the average risk must decrease over time. Combining this with Theorem 2.2, GD must enter the stable phase in finite steps, and the loss must converge while the normalized margin must improve.

However, a direct application of Theorem 3.2 only leads to a suboptimal bound on the phase transition time. Motivated by Wu et al. [2024], we establish the following sharp bound on the phase transition time by tracking the gradient potential (see Lemma C.3). The proof is deferred to Appendix C.

**Theorem 4.1** (Phase transition and stable phase for two-layer networks). *Under Assumption 3, consider* (GD) *on two-layer networks* (2) *that satisfy Assumption 2. Clearly, the two-layer networks also satisfy Assumption 1 with* $\rho = 1/\sqrt{m}$, $\beta = \tilde{\beta}/m$, *and* $\kappa = \kappa$. *Denote the stepsize by* $\tilde{\eta} := m\eta$, *where* $m$ *is the network width and* $\eta > 0$ *can be arbitrarily large.*

- **Phase transition time**. There exists $s \leq \tau$ such that (5) in Theorem 2.2 holds, where

$$\tau := \frac{128(1 + 4\kappa)}{\alpha^2} \max\left\{ c_1\eta, c_2 n, e, \frac{c_2\eta + c_1 n}{\eta} \log \frac{c_2\eta + c_1 n}{\eta}, (c_2\eta + c_1 n) \cdot \frac{\|\mathbf{w}_0\|}{\sqrt{m}} \right\},$$

where $c_1 := 4e^{\kappa+2}$ and $c_2 := (8 + 4\tilde{\beta})$. Therefore (GD) is in the stable phase from $s$ onwards.

- **Explicit risk bound in the stable phase**. We have $(L(\mathbf{w}_t))_{t \geq s}$ monotonically decreases and

$$L(\mathbf{w}_t) \leq \frac{2}{\alpha^2\gamma^2\eta(t - s)}, \quad t \geq s.$$

Theorems 2.2, 3.2 and 4.1 together characterize the behaviors of large stepsize GD in training two-layer networks. Specifically, large stepsize GD may induce an oscillatory risk in the beginning; but the averaged empirical risk must decrease (Theorem 3.2). After the empirical risk falls below a certain stepsize-dependent threshold, GD enters the stable phase, where the risk decreases monotonically (Theorem 4.1). Finally, the normalized margin (3) induced by GD increases nearly monotonically as GD stays in the stable phase (Theorem 2.2).

Our intuition behind the phase transition phenomenon is as follows. The initial EoS phase occurs when gGD oscillates within a steep valley, transitioning to a stable phase once it navigates into a flatter valley. We believe this insight generalizes to broader nonlinear models. Moreover, our theory of large step sizes aligns with the celebrated flat minima intuition [Keskar et al., 2016].

**Fast optimization.**   Our bounds for two-layer networks are comparable to those for linear predictors shown by Wu et al. [2024]. Specifically, when used with a larger stepsize, GD achieves a faster optimization in the stable phase but stays longer in the EoS phase. Choosing a suitably large stepsize that balances the steps in EoS and stable phases, we obtain an *accelerated* empirical risk in the following corollary. The proof is included in Appendix C.2.

**Corollary 4.2** (Acceleration of large stepsize). *Under the same setup as in Theorem 4.1, consider* (GD) *with a given budget of $T$ steps such that*

$$T \geq \frac{256(1 + 4\kappa)}{\alpha^2\gamma^2} \max\left\{ c_1 n, \ 4c_2^2, \ \frac{2c_2\|\mathbf{w}_0\|}{\sqrt{m}} \right\},$$

*where $c_1 := 4e^{\kappa+2}$ and $c_2 := (8 + 4\tilde{\beta})$ are as in Theorem 4.1. Then for stepsize $\tilde{\eta} := \eta m$, where*

$$\eta := \frac{\alpha^2\gamma^2}{256(1 + 4\kappa)c_2}T,$$

*we have $\tau \leq T/2$ and*

$$L(\mathbf{w}_T) \leq \frac{2048(1 + 4\kappa)c_2}{\alpha^4\gamma^4} \cdot \frac{1}{T^2} = \mathcal{O}\left(\frac{1}{T^2}\right).$$

Theorem 4.1 and Corollary 4.2 extend Theorem 1 and Corollary 2 in Wu et al. [2024] from linear predictors to two-layer networks. Another notable difference is that we obtain a sharper stable phase bound (and thus a better acceleration bound) compared to theirs, where we remove a logarithmic factor through a more careful analysis.

Corollary 4.2 suggests an accelerated risk bound of $\mathcal{O}(1/T^2)$ by choosing a large stepsize that balances EosS and stable phases. We also show the following lower bound, showing that such acceleration is impossible if (GD) does not enter the EoS phase. The proof is included in Appendix C.3.

**Theorem 4.3** (Lower bound in the classical regime). *Consider* (GD) *with initialization $\mathbf{w}_0 = 0$ and stepsize $\tilde{\eta} > 0$ for a two-layer network* (2) *satisfying Assumption 2. Suppose the training set is given by*

$$\mathbf{x}_1 = (\gamma, \sqrt{1 - \gamma^2}), \quad \mathbf{x}_2 = (\gamma, -\sqrt{1 - \gamma^2}/2), \quad y_1 = y_2 = 1, \quad 0 < \gamma < 0.1.$$

*It is clear that $(\mathbf{x}_i, y_i)_{i=1,2}$ satisfy Assumption 3. If $(L(\mathbf{w}_t))_{t \geq 0}$ is non-increasing, then*

$$L(\mathbf{w}_t) \geq c_0/t, \quad t \geq 1$$

*where $c_0 > 0$ is a function of $(\alpha, \phi, \mathbf{x}_1, \mathbf{x}_2, \gamma, \kappa, \beta)$ but is independent of $t$ and $\tilde{\eta}$.*

**Effect of model rescaling.** We conclude this section by discussing the impact of rescaling the model. Specifically, we replace the two-layer network in the mean-field scaling (2) by the following

$$f(\mathbf{w}; \mathbf{x}) := b \cdot \frac{1}{m} \sum_{j=1}^{m} a_j \phi(\mathbf{x}^\top \mathbf{w}^{(j)}),$$

and evaluate the impact of the scaling factor $b$ on our results. By choosing the optimal stepsize that balances the EoS and stable phases as in Corollary 4.2, we optimize the risk bound obtained by GD with a fixed budget of $T$ steps and get the following bound. Detailed derivations are deferred to Appendix D.

$$L(\mathbf{w}_T) = \begin{cases} \mathcal{O}(1/T^2) & \text{if } b \geq 1, \\ \mathcal{O}(b^{-3}/T^2) & \text{if } b < 1. \end{cases}$$

This suggests that as long as $b \geq 1$, we get the same acceleration effect. In particular, the mean-field scaling $b = 1$ [Song et al., 2018, Chizat and Bach, 2020] and the *neural tangent kernel* (NTK) scaling $b = \sqrt{m}$ [Du et al., 2018, Jacot et al., 2018] give the same acceleration effect. An NTK analysis of large stepsize is included in [Wu et al., 2024] and their conclusion is consistent with ours. Finally, we remark that our analysis holds for any width $m$ and uses techniques different from the mean-field or NTK methods. However, our acceleration analysis only allows linearly separable datasets.

## 5  Experiments

We conduct three sets of experiments to validate our theoretical insights. In the first set, we use a subset of the CIFAR-10 dataset [Krizhevsky et al., 2009], which includes 6,000 randomly selected samples from the "airplane" and "automobile" classes. Our model is a multilayer perceptron (MLP) with four trainable layers and GELU activation functions, with a hidden dimension of 200 for each hidden layer. The MLP is trained using gradient descent with random initialization, as described in (GD). The results are shown in Figures 1(a) to 1(c).

In the second set of experiments, we consider an XOR dataset consisting of four samples:

$$\mathbf{x}_1 = (-1, -1), y_1 = 1; \ \mathbf{x}_2 = (1, 1), y_2 = 1; \ \mathbf{x}_3 = (1, -1), y_3 = -1; \ \mathbf{x}_4 = (-1, 1), y_4 = -1.$$

The above XOR dataset is not linearly separable. We test (GD) with different stepsizes on a two-layer network (2) with the leaky softplus activation (see Example 3.1 with $c = 0.5$). The network width is $m = 20$. The initialization is random. The results are presented in Figures 2(a) to 2(c).

In the third set of experiments, we consider the same task as in the first set of experiments, but we test (GD) with different stepsizes on a two-layer network (2) with the softplus activation. The network width is $m = 40$. The initialization is random. The results are presented in Figures 2(d) to 2(f).

**Margin improvement.** Figures 1(b), 2(c) and 2(f) show that the normalized margin nearly monotonically increases once gradient descent (GD) enters the stable phase, regardless of step size. This observation aligns with our theoretical findings in Theorem 2.2.

**Fast optimization.** From Figures 1(a), 2(a) and 2(d), we observe that after GD enters the stable phase, a larger stepsize consistently leads to a smaller empirical risk compared to the smaller stepsizes, which is consistent with our Theorem 4.1 and Corollary 4.2. Besides, Figures 2(b) and 2(e) suggest that, asymptotically, GD converges at a rate of $\mathcal{O}(1/(\tilde{\eta}t)) = \mathcal{O}(1/(\eta t))$ (The width of networks is fixed), which verifies the sharpness of our stable phase bound in Theorem 4.1.

**Margin of individual neurons.** It is important to note that while the normalized margin behaves as expected, the margin for individual neurons may not increase and can remain negative, even when the dataset is linearly separable. A detailed example illustrating this is provided in Appendix E.

## 6  Related Works

In this section, we discuss related papers.

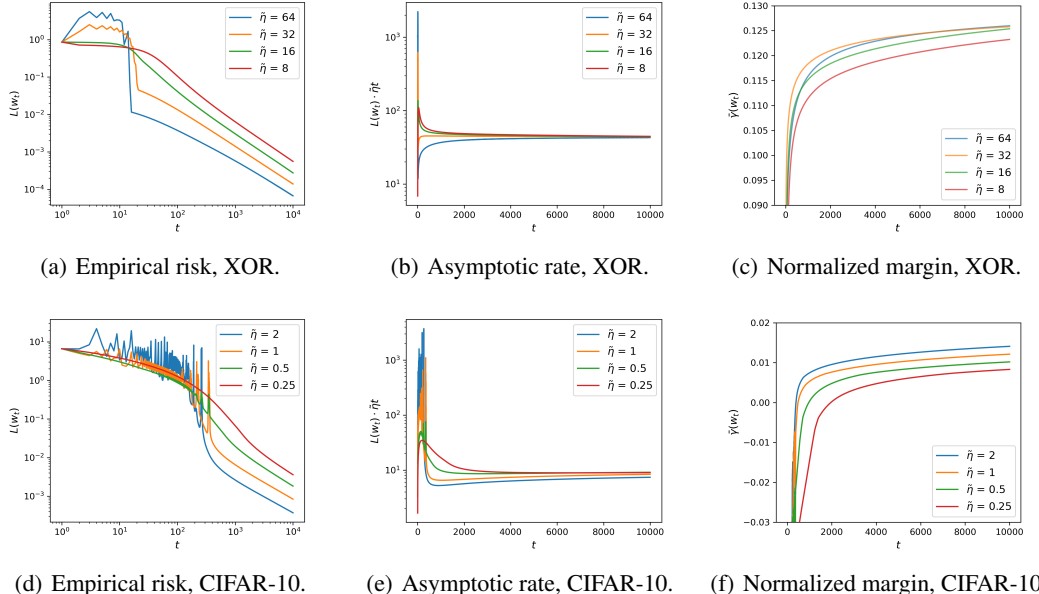

(a) Empirical risk, XOR.   (b) Asymptotic rate, XOR.   (c) Normalized margin, XOR.

(d) Empirical risk, CIFAR-10.  (e) Asymptotic rate, CIFAR-10.  (f) Normalized margin, CIFAR-10.

Figure 2: Behavior of (GD) for two-layer networks (2) with leaky softplus activation function (see Example 3.1 with $c = 0.5$). We consider an XOR dataset and a subset of CIFAR-10 dataset. In both cases, we observe that (1) GD with a large stepsize achieves a faster optimization compared to GD with a small stepsize, (2) the asymptotic convergence rate of the empirical risk is $\mathcal{O}(1/(\tilde{\eta}t))$, and (3) in the stable phase, the normalized margin (nearly) monotonically increases. These observations are consistent with our theoretical understanding of large stepsize GD. More details about the experiments are explained in Section 5.

**Small stepsize and implicit bias.** For logistic regression on linearly separable data, Soudry et al. [2018], Ji and Telgarsky [2018] showed that the direction of small stepsize GD converges to the max-margin solution. Their results were later extended by Gunasekar et al. [2017, 2018], Nacson et al. [2019c,a,b], Ji et al. [2021], Lyu and Li [2020], Ji and Telgarsky [2020], Chizat and Bach [2020], Chatterji et al. [2021], Kunin et al. [2022] to other algorithms and non-linear models. However, in all of their analysis, the stepsize of GD needs to be small such that the empirical risk decreases monotonically. In contrast, our focus is GD with a large stepsize that induces non-monotonic risk.

Two papers [Nacson et al., 2019a, Kunin et al., 2022] studied margin maximization theory for a special form of non-homogenous models. Specifically, when viewed in terms of different subsets of the trainable parameters, the model is homogeneous, although the order of homogeneity may vary. Compared to their setting, our non-homogenous models only require a bounded homogenous error (see Assumption 1C). Therefore, our theory can cover two-layer networks (2) with non-homogeneous activations such as GELU and SiLU that cannot be covered by [Nacson et al., 2019a, Kunin et al., 2022].

**Large stepsize and EoS.** In practice, large stepsizes are often preferred when using GD to train neural networks to achieve effective optimization and generalization performance [see Wu and Ma, 2018, Cohen et al., 2020, Barrett and Dherin, 2020, and references therein]. In such scenarios, the empirical risk often oscillates in the beginning. This phenomenon is named *edge of stability* (EoS) by Cohen et al. [2020]. The theory of EoS is mainly studied in relatively simplified cases such as one-or two-dimensional functions [Zhu et al., 2022, Chen and Bruna, 2023, Ahn et al., 2022, Kreisler et al., 2023, Wang et al., 2023], linear model [Wu et al., 2023, 2024], matrix factorization [Wang et al., 2022a, Chen and Bruna, 2023], scale-invariant networks [Lyu et al., 2022], linear networks under MSE loss [Ren et al., 2024, Andriushchenko et al., 2023], for an incomplete list of references. Compared to them, we focus on a more practical setup of training two-layer non-linear networks with large stepsize GD. There are some general theories of EoS subject to subtle assumptions [for

example, Kong and Tao, 2020, Ahn et al., 2022, Ma et al., 2022, Damian et al., 2022, Wang et al., 2022b, Lu et al., 2023], which are not directly comparable to ours.

In what follows, we make a detailed discussion about papers that directly motivate our work [Lyu and Li, 2020, Ji and Telgarsky, 2020, Chatterji et al., 2021, Wu et al., 2024].

**Comparison with Lyu and Li [2020], Ji and Telgarsky [2020].** Both results in [Lyu and Li, 2020, Ji and Telgarsky, 2020] focused on $L$-homogenous networks. Specifically, Lyu and Li [2020] showed that a modified version of normalized margin (see (3)) induced by GD with small stepsize (such that the risk decreases monotonically) increases, with limiting points of $\{\mathbf{w}_t/\|\mathbf{w}_t\|\}_{t=1}^{\infty}$ converging to KKT points of a margin-maximization problem. Under additional o-minimal conditions, Ji and Telgarsky [2020] showed that gradient flow converges in direction. Our work is different from theirs in two aspects. First, we allow GD with a large stepsize that may cause risk oscillation. Second, our theory covers non-homogenous predictors, which include two-layer networks with many commonly used activation functions beyond the scope of [Lyu and Li, 2020, Ji and Telgarsky, 2020]. Compared to Lyu and Li [2020], Ji and Telgarsky [2020], we only show the improvement of the margin, and our theory is limited to nearly 1-homogenous predictors (Assumption 2C). It remains open to show directional convergence and to extend our near 1-homogenity condition to a "near $L$-homogeneity" condition for a general $L$.

**Comparison with Chatterji et al. [2021].** The work by Chatterji et al. [2021] studies the convergence of GD in training deep networks under logistic loss. Their results are related to ours as we both consider networks with nearly homogeneous activations and we both have a stable phase analysis (although this is not explicitly mentioned in their paper). However, our results are significantly different from theirs. Specifically, in our notation, they require the homogenous error $\kappa$ (see Assumption 2C) to be smaller than $\mathcal{O}(\log(1/L(\mathbf{w}_s))/\|\mathbf{w}_s\|) \approx \mathcal{O}(\bar{\gamma}(\mathbf{w}_s))$, where $s$ is the time for GD to enter the stable phase. Note that the margin when GD enters the stable phase could be arbitrarily small. In comparison, we only require the homogenous error to be bounded by a constant. As a consequence, we can handle many commonly used activation functions (see Example 2.1) while they can only handle the Huberized ReLU with a small $h$ in Example 2.1. Moreover, they require the stepsize $\tilde{\eta}$ to be smaller than $\mathcal{O}(\kappa/\|\mathbf{w}_s\|^8)$, thus they only allow very small stepsize. In contrast, we allow $\tilde{\eta}$ to be arbitrarily large.

**Comparison with Wu et al. [2023, 2024].** The works by Wu et al. [2023, 2024] directly motivate our paper. In particular, for logistic regression on linearly separable data, Wu et al. [2023] showed margin maximization of GD with large stepsize and Wu et al. [2024] showed fast optimization of GD with large stepsize. Our work can be viewed as an extension of [Wu et al., 2023, 2024] from linear predictors to non-linear predictors such as two-layer networks. Besides, our results for margin improvement and convergence within the stable phase (Theorem 2.2) hold for the general dataset, while their results strongly rely on the linear separability of the dataset.

## 7 Conclusion

We provide a theory of large stepsize gradient descent (GD) for training non-homogeneous predictors such as two-layer networks using the logistic loss function. Our analysis explains the empirical observations: large stepsize GD often reveals two distinct phases in the training process, where the empirical risk oscillates in the beginning but decreases monotonically subsequently. We show that the phase transition happens because the average empirical risk decreases despite the risk oscillation. In addition, we show that large stepsize GD improves the normalized margin in the long run, which extends the existing implicit bias theory for homogenous predictors to non-homogenous predictors. Finally, we show that large stepsize GD, by entering the initial oscillatory phase, achieves acceleration when minimizing the empirical risk.

## Acknowledgements

We thank Fabian Pedregosa for his suggestions on an early draft and Jason D. Lee and Kaifeng Lyu for their comments clarifying the applicability of our result to sigmoid networks. We gratefully acknowledge the support of the NSF for FODSI through grant DMS-2023505, of the NSF and the

Simons Foundation for the Collaboration on the Theoretical Foundations of Deep Learning through awards DMS-2031883 and #814639, and of the ONR through MURI award N000142112431.

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

# A    Stable Phase Analysis

In this section, we will prove results for a general smooth predictor $f(\mathbf{w}; \mathbf{x})$ under the logistic loss in the stable phase. Before the proof, we introduce some notations here.

**Notation.**    We use the following notation to simplify the presentation.

- $q_i(t) := y_i f(\mathbf{w}_t; \mathbf{x}_i),\ q_{\min}(t) := \min_{i \in [n]} q_i(t)$.
- $L_t := L(\mathbf{w}_t),\ \rho_t := \|\mathbf{w}_t\|_2$.

Then, we have the following expression:

$$L(\mathbf{w}_t) = \frac{1}{n} \sum_{i=1}^{n} \ell(q_i(t)).$$

Here, we give a summary of this section. The proofs are organized into 5 parts.

- In Appendix A.1, we characterize the decrease of loss $L_t$.
- In Appendix A.2, we characterize the change of the parameter norm $\rho_t$.
- In Appendix A.3, we show the convergence of the normalized margin $\bar{\gamma}(\mathbf{w}_t)$.
- In Appendix A.4, we characterize the sharp rates of loss $L_t$ and parameter norm $\rho_t$.
- In Appendix A.5, we give the proof of Theorem 2.2.

## A.1    Decrease of the Loss

In this section, we will show that the loss $L_t$ decreases monotonically in the stable phase. To begin with, we introduce the following definition which is another characterization of $\beta$-smoothness.

**Definition 1** (Linearization error). Given a continuously differentiable function $f : \mathbb{R}^d \to \mathbb{R}$ and two points $\mathbf{w}, \mathbf{v} \in \mathbb{R}^d$, the linearization error of $f(\mathbf{v})$ with respect to $\mathbf{w}$ is:

$$\xi[f](\mathbf{w}, \mathbf{v}) := f(\mathbf{v}) - f(\mathbf{w}) - \nabla f(\mathbf{w})^\top (\mathbf{v} - \mathbf{w}).$$

For a $\beta$-smooth function, standard convex optimization theory gives the following linearization error bound.

**Fact A.1** (Linearization error of $\beta$-smooth function). *For a $\beta$-smooth function $f : \mathbb{R}^d \to \mathbb{R}$, we have*

$$\xi[f](\mathbf{w}, \mathbf{v}) := f(\mathbf{v}) - f(\mathbf{w}) - \nabla f(\mathbf{w})^\top (\mathbf{v} - \mathbf{w}) \leq \frac{\beta}{2} \|\mathbf{v} - \mathbf{w}\|_2^2, \quad \textit{for every } \mathbf{w} \textit{ and } \mathbf{v}.$$

We first show a stable phase bound for general smooth and Lipschitz predictors. The following is an extension of Lemma 10 in [Wu et al., 2024]. Since we do not require $f$ to be twice differentiable, extra efforts are needed.

**Lemma A.2** (Self-boundedness of logistic loss). *For the logistic loss $\ell(z) := \log(1 + \exp(-z))$, we have*

$$0 \leq \ell(z) - \ell(x) - \ell'(x)(z - x) \leq 2\ell(x)(z - x)^2$$

*for $|z - x| < 1$.*

*Proof of Lemma A.2.* See the proof of Proposition 5 in [Wu et al., 2024]. The lower bound is by the convexity of $\ell(\cdot)$. $\square$

The next lemma controls the decrease of the risk $L_t$.

**Lemma A.3** (Decrease of $L_t$). *Suppose Assumptions 1A and 1B hold. If $L(\mathbf{w}_t) \leq \frac{1}{\tilde{\eta}\rho^2}$, then we have*

$$-\tilde{\eta}(1 + \beta\tilde{\eta}L(\mathbf{w}_t))\|\nabla L(\mathbf{w}_t)\|^2 \leq L(\mathbf{w}_{t+1}) - L(\mathbf{w}_t) \leq -\tilde{\eta}(1 - (2\rho^2 + \beta)\tilde{\eta}L(\mathbf{w}_t))\|\nabla L(\mathbf{w}_t)\|^2.$$

*Particularly, this indicates that if $L(\mathbf{w}_t) \leq \frac{1}{\tilde{\eta}(2\rho^2+\beta)}$, then $L(\mathbf{w}_{t+1}) \leq L(\mathbf{w}_t)$.*

*Proof of Lemma A.3.* By Assumptions 1A and 1B, we have $\|\nabla f\|_2 \le \rho$ and $f(\mathbf{w}; \mathbf{x})$ is $\beta$-smooth as a function of $\mathbf{w}$. Therefore, for every $i \in [n]$ we have

$$
\begin{aligned}
|q_i(t+1) - q_i(t)| &= |y_i(f(\mathbf{w}_{t+1}; \mathbf{x}_i) - f(\mathbf{w}_t; \mathbf{x}_i))| \\
&= \left| \nabla f(\mathbf{w}_t + \theta(\mathbf{w}_{t+1} - \mathbf{w}_t); \mathbf{x}_i)^\top (\mathbf{w}_{t+1} - \mathbf{w}_t) \right| \quad \text{by intermediate value theorem} \\
&\le \rho \|\mathbf{w}_{t+1} - \mathbf{w}_t\| \\
&\le \rho \tilde{\eta} \|\nabla L_t\| \quad\quad\quad\quad\quad\quad\quad\quad\quad\quad \text{since } \mathbf{w}_{t+1} = \mathbf{w}_t - \tilde{\eta}\nabla L_t \\
&\le \rho^2 \tilde{\eta} L_t \le 1. \quad\quad\quad\quad\quad\quad\quad\quad\quad \text{since } \|\nabla L_t\| \le L_t \rho
\end{aligned}
$$

Then by Lemma A.2, we have

$$
\begin{aligned}
\ell(q_i(t+1)) &\le \ell(q_i(t)) + \ell'(q_i(t))(q_i(t+1) - q_i(t)) + 2\ell(q_i(t))(q_i(t+1) - q_i(t))^2 \\
&\le \ell(q_i(t)) + \ell'(q_i(t))\langle y_i \nabla f(\mathbf{w}_t; \mathbf{x}_i), \mathbf{w}_{t+1} - \mathbf{w}_t \rangle + |\ell'(q_i(t))| \cdot |\xi[f](\mathbf{w}_t, \mathbf{w}_{t+1})| \\
&\quad + 2\ell(q_i(t))(q_i(t+1) - q_i(t))^2 \\
&\quad \text{since } q_i(t+1) - q_i(t) = \langle y_i \nabla f(\mathbf{w}_t; \mathbf{x}_i), \mathbf{w}_{t+1} - \mathbf{w}_t \rangle + y_i \xi[f](\mathbf{w}_t, \mathbf{w}_{t+1}) \\
&\le \ell(q_i(t)) + \ell'(q_i(t))\langle y_i \nabla f(\mathbf{w}_t; \mathbf{x}_i), \mathbf{w}_{t+1} - \mathbf{w}_t \rangle + \ell(q_i(t))(\beta + 2\rho^2)\|\mathbf{w}_{t+1} - \mathbf{w}_t\|^2. \\
&\quad \text{by Fact A.1 and the previous inequality}
\end{aligned}
$$

Taking an average over all data points, we have

$$
L_{t+1} \le L_t - \tilde{\eta}\|\nabla L_t\|^2 + (2\rho^2 + \beta)\tilde{\eta}^2 L_t \|\nabla L_t\|^2,
$$

which is equivalent to

$$
L_{t+1} - L_t \le -\tilde{\eta}(1 - (2\rho^2 + \beta)\tilde{\eta} L_t)\|\nabla L_t\|^2.
$$

We complete the proof of the right hand side inequality. The left hand side inequality can be proved similarly. In detail, we can show that:

$$
\begin{aligned}
\ell(q_i(t+1)) &\ge \ell(q_i(t)) + \ell'(q_i(t))(q_i(t+1) - q_i(t) \\
&\ge \ell(q_i(t)) + \ell'(q_i(t))\langle y_i \nabla f(\mathbf{w}_t; \mathbf{x}_i), \mathbf{w}_{t+1} - \mathbf{w}_t \rangle - |\ell'(q_i(t))| \cdot |\xi[f](\mathbf{w}_t, \mathbf{w}_{t+1})|.
\end{aligned}
$$

Taking the average over all data points, we have

$$
L_{t+1} \ge L_t - \tilde{\eta}(1 + \beta \tilde{\eta} L_t)\|\nabla L_t\|^2.
$$

Now we have completed the proof. $\qquad \square$

## A.2 Increase of the Parameter Norm

In this section, we demonstrate that the parameter norm, $\rho_t$, increases monotonically during the stable phase. We introduce a crucial quantity, $v_t$, defined as the inner product of the gradient and the negative weight vector:

$$
v_t := \langle \nabla L(\mathbf{w}_t), -\mathbf{w}_t \rangle.
$$

This quantity, $v_t$, plays a key role in controlling the increase of the parameter norm. Notably, $v_t$ appears as the cross term in the expression $\|\mathbf{w}_{t+1}\|^2 = \|\mathbf{w}_t - \tilde{\eta}\nabla L(\mathbf{w}_t)\|^2$. By managing $v_t$, we can effectively characterize the increase in the parameter norm.

Recall that our loss function is $\ell(x) := \log(1 + e^{-x})$. Inspired by Lyu and Li [2020], we define the following two auxiliary functions for the logistic loss:

$$
\begin{aligned}
\psi(x) &:= -\log(\ell(x)) = -\log\log(1 + e^{-x}), \quad x \in \mathbb{R}, \\
\iota(x) &:= \psi^{-1}(x) = -\log(e^{e^{-x}} - 1), \quad x \in \mathbb{R}.
\end{aligned} \tag{6}
$$

One important remark is that if we change the loss to the exponential loss, both $\psi$ and $\iota$ will be the identity function. Since the logistic loss and the exponential loss have similar tails, our $\psi(x)$ and $\iota(x)$ are close to the identity function, i.e.,

$$
\psi(x) \approx \iota(x) \approx x, \quad \text{for } x \text{ large enough.}
$$

Then, we have an exponential-loss-like decomposition of $L_t$:

$$
L_t = \frac{1}{n}\sum_{i=1}^n \ell(q_i(t)) = \frac{1}{n}\sum_{i=1}^n e^{-\psi(q_i(t))}. \tag{7}
$$

These two functions $\psi, \iota$ will help us to analyze the lower bound of $v_t$. First, we list some properties of $\psi$ and $\iota$ here.

**Lemma A.4** (Auxiliary functions of $\ell$). *The following claims hold for $\ell$, $\psi$, and $\iota$.*

- $\ell(x) = e^{-\psi(x)}$.

- *$\ell$ is monotonically decreasing, while $\psi$ and $\iota$ are monotonically increasing.*

- $\psi'(\iota(x)) = \frac{1}{\iota'(x)}$;

- *$\psi'(x)x$ is increasing for $x \in (0, +\infty)$.*

*Proof of Lemma A.4.* The first two properties are straightforward. For the third property, we apply chain rule on $\psi(\iota(x)) = x$ to get

$$\psi'(\iota(x))\iota'(x) = 1.$$

For the fourth property, notice that

$$\psi'(x)x = \frac{x}{(1 + e^x)\log(1 + e^{-x})}.$$

The denominator is positive and decreasing since

$$\frac{d}{dx}\left[(1 + e^x)\log(1 + e^{-x})\right] = e^x \log(1 + e^{-x}) - 1 \le e^x e^{-x} - 1 = 0.$$

Combining this with the fact that $x$ is positive and increasing, we have the desired result. $\square$

Besides, we have the following property of $\iota$. This is the key lemma to handle the homogeneous error. Actually, this lemma is another way to show $\iota(x)$ is close to the identity function.

**Lemma A.5** (Property of $\iota$). *For every $x \in \mathbb{R}$, we have*

$$\frac{\iota(x)}{\iota'(x)} \ge x + \log \log 2.$$

*Proof of Lemma A.5.* Recall that

$$\iota(x) = -\log(e^{e^{-x}} - 1), \quad \iota'(x) = \frac{e^{e^{-x}}e^{-x}}{e^{e^{-x}} - 1}.$$

Let $y = e^{-x}$. We have

$$\frac{\iota(x)}{\iota'(x)} = \frac{-\log(e^{e^{-x}} - 1)(e^{e^{-x}} - 1)}{e^{e^{-x}}e^{-x}} = \frac{-\log(e^y - 1)(e^y - 1)}{e^y y}.$$

Define $s(y) := \frac{\iota(x)}{\iota'(x)} - x - \log \log 2$. Then,

$$s(y) = \frac{-\log(e^y - 1)(e^y - 1)}{e^y y} + \log(y) - \log \log 2,$$

$$s'(y) = \log(e^y - 1) \cdot \underbrace{\frac{e^y - y - 1}{e^y y^2}}_{>0}.$$

Note that the sign of $s'$ is determined by $\log(e^y - 1)$. For $0 < e^y \le 2$, $s'(y) \le 0$ and $s(y)$ is decreasing; for $e^y \ge 2$, $s(y)$ is increasing. Therefore,

$$\min_{y \in (0,\infty)} s(y) = s(\log 2) = 0.$$

Since $x = -\log y$, we have the desired result. $\square$

Another important property of $\iota$ is that it can provide a lower bound for $q_{\min}(t)$.

**Lemma A.6** ($\iota$ bound $q_{\min}$). *For every $t \ge 0$, we have*

$$q_{\min}(t) \ge \iota\left(-\log(L_t) - \log n\right).$$

*Proof of Lemma A.6.* We use (7) to get

$$\frac{1}{n}\ell(q_{\min}(t)) \leq L_t \Rightarrow \frac{1}{n}e^{-\psi(q_{\min}(t))} \leq L_t$$
$$\Rightarrow \psi(q_{\min}(t)) \geq -\log n - \log L_t$$
$$\Rightarrow q_{\min}(t) \geq \iota\big(-\log(L_t) - \log n\big). \qquad \text{by Lemma A.4}$$

Then, we complete the proof. $\qquad\square$

Now, we are ready to give a lower bound of $v_t$. The following lemma is an extension of Corollary E.6 in Lyu and Li [2020], where they dealt with a homogeneous model and the exponential loss; we extend this to a non-homogeneous model. The key ingredient is Lemma A.5.

**Lemma A.7** (A lower bound of $v_t$). *Suppose Assumption 1C holds. Consider $v_t := \langle \nabla L(\mathbf{w}_t), -\mathbf{w}_t \rangle$. If $L_t \leq \frac{1}{2ne^{\kappa}}$, then*

$$v_t \geq -L_t \log(2ne^{\kappa}L_t) \geq 0.$$

*Proof of Lemma A.7.* By definition, we have

$$v_t := \langle \nabla L(\mathbf{w}_t), -\mathbf{w}_t \rangle$$
$$= -\frac{1}{n}\sum_{i=1}^{n}\ell'(y_i f(\mathbf{w}_t; \mathbf{x}_i))y_i\langle \nabla f(\mathbf{w}_t; \mathbf{x}_i), \mathbf{w}_t \rangle$$
$$= -\frac{1}{n}\sum_{i=1}^{n}\ell'(y_i f(\mathbf{w}_t; \mathbf{x}_i))y_i f(\mathbf{w}_t; \mathbf{x}_i) - \frac{1}{n}\sum_{i=1}^{n}\ell'(y_i f(\mathbf{w}_t; \mathbf{x}_i))\Big(y_i\langle \nabla f(\mathbf{w}_t; \mathbf{x}_i), \mathbf{w}_t \rangle - y_i f(\mathbf{w}_t; \mathbf{x}_i)\Big)$$
$$\geq -\frac{1}{n}\sum_{i=1}^{n}\ell'(y_i f(\mathbf{w}_t; \mathbf{x}_i))y_i f(\mathbf{w}_t; \mathbf{x}_i) - \kappa L_t$$

    since $|\ell'(x)| \leq \ell(x)$ and $|\langle \nabla f(\mathbf{w}_t; \mathbf{x}_i), \mathbf{w}_t \rangle - f(\mathbf{w}_t; \mathbf{x}_i)| \leq \kappa$ by Assumption 1C

$$= \frac{1}{n}\sum_{i=1}^{n}e^{-\psi(q_i(t))}\psi'(q_i(t))q_i(t) - \kappa L_t. \qquad \text{since } \ell(\cdot) = \exp(-\psi(\cdot))$$

Applying Lemma A.6 and Lemma A.4, we have

$$q_i(t) \geq q_{\min}(t) \geq \iota\big(-\log(nL_t)\big) := -\log(e^{nL_t} - 1) \geq -\log(e^{\frac{1}{2}} - 1) \geq 0.$$

Then we can apply Lemma A.4 to get

$$\psi'(q_i(t))q_i(t) \geq \psi'\Big(\iota\big(-\log(nL_t)\big)\Big)\iota\big(-\log(nL_t)\big) = \frac{\iota\big(-\log(nL_t)\big)}{\iota'\big(-\log(nL_t)\big)}.$$

Invoking Lemma A.5, we have

$$\frac{\iota\big(-\log(nL_t)\big)}{\iota'\big(-\log(nL_t)\big)} \geq -\log(nL_t) + \log\log 2 \geq -\log(nL_t) + \log\log e^{\frac{1}{2}} = -\log(2nL_t).$$

Putting the above two inequalities together, we have

$$\psi'(q_i(t))q_i(t) \geq -\log(2nL_t), \quad \text{for every } i = 1, \ldots, n.$$

Plugging this back to the bound of $v_t$, we get

$$v_t \geq -\frac{1}{n}\sum_{i=1}^{n}e^{-\psi(q_i(t))}\log(2nL_t) - \kappa L_t$$
$$= -L_t\log(2nL_t) - \kappa L_t$$
$$= -L_t\log(2ne^{\kappa}L_t) \geq 0.$$

This completes the proof. $\qquad\square$

Right now, we get a lower bound for $v_t$, which is the cross term in the expression of $\|\mathbf{w}_{t+1}\|^2$. The next lemma controls the increase of the parameter norm $\rho_t$ using $v_t$ and $L_t$.

**Lemma A.8** (The increase of $\rho_t$). *Suppose Assumptions 1A and 1C hold. If $L_t \leq \min\left\{\frac{1}{2ne^{\kappa}}, \frac{1}{\tilde{\eta}(4\rho^2 + 2\beta)}\right\}$, then*

$$0 \leq 2\tilde{\eta}v_t \leq \rho_{t+1}^2 - \rho_t^2 \leq 2\tilde{\eta}v_t \cdot \left(1 - \frac{1}{2\log(2ne^{\kappa}L_t)}\right).$$

*Proof of Lemma A.8.* By definition, we have

$$\rho_{t+1}^2 - \rho_t^2 = 2\tilde{\eta}\langle \nabla L_t, -\mathbf{w}_t \rangle + \tilde{\eta}^2 \|\nabla L_t\|^2$$
$$= 2\tilde{\eta}v_t + \tilde{\eta}^2\|\nabla L_t\|^2 \geq 2\tilde{\eta}v_t \geq 0,$$

where the last inequality is by Lemma A.7. Besides,

$$\rho_{t+1}^2 - \rho_t^2 = 2\tilde{\eta}v_t\left(1 + \frac{\tilde{\eta}\|\nabla L_t\|^2}{2v_t}\right)$$
$$\leq 2\tilde{\eta}v_t\left(1 + \frac{\tilde{\eta}L_t^2\rho^2}{2v_t}\right) \qquad \text{by } \ell' \leq \ell, \text{ Assumption 1A, and Lemma A.7}$$
$$\leq 2\tilde{\eta}v_t\left(1 + \frac{L_t}{2v_t}\right) \qquad \text{by } L_t \leq \frac{1}{\tilde{\eta}(4\rho^2 + 2\beta)}$$
$$\leq 2\tilde{\eta}v_t\left(1 - \frac{1}{2\log(2ne^{\kappa}L_t)}\right). \qquad \text{by Lemma A.7}$$

This completes the proof. $\qquad \square$

### A.3 Convergence of the Margin

In this section, we show that the normalized margin of a general predictor converges in the stable phase. Recall that we define the (normalized) margin as

$$\bar{\gamma}(\mathbf{w}) := \frac{\min_{i \in [n]} y_i f(\mathbf{w}; \mathbf{x}_i)}{\|\mathbf{w}\|_2}.$$

However, this normalized margin is not a smooth function of $\mathbf{w}$. Instead, we consider a smoothed margin $\gamma^a$ as an easy-to-analyze approximator of the normalized margin [Lyu and Li, 2020]

$$\gamma^a(\mathbf{w}) := \frac{-\log L(\mathbf{w}_t)}{\|\mathbf{w}\|_2}. \qquad (8)$$

We see that $\gamma^a$ is a good approximator of $\bar{\gamma}$. We can then use $\gamma^a$ to analyze the convergence of the normalized margin since they share the same limit (if it exists). While $\gamma^a$ is relatively easy to analyze for gradient flow [Lyu and Li, 2020], analyzing that for GD with a large (but fixed) stepsize is hard. To mitigate this issue, we construct another two margins that work well with large stepsize GD following the ideas of Lyu and Li [2020].

Under Assumption 1, we define an auxiliary margin as

$$\gamma^b(\mathbf{w}) := \frac{-\log(2ne^{\kappa}L(\mathbf{w}))}{\|\mathbf{w}\|}, \qquad (9)$$

and a modified margin as

$$\gamma^c(\mathbf{w}) := \frac{e^{\Phi(L(\mathbf{w}))}}{\|\mathbf{w}\|}, \quad \text{where } \Phi(x) := \log(-\log(2ne^{\kappa}x)) + \frac{1 + (4\rho^2 + 2\beta)\tilde{\eta}}{\log(2ne^{\kappa}x)}. \qquad (10)$$

These two margins provide a second-order correction when viewing large stepsize GD as a first-order approximation of gradient flow. In the following discussion, we will show that $\bar{\gamma}(\mathbf{w}) \approx \gamma^a(\mathbf{w}) \approx \gamma^b(\mathbf{w}) \approx \gamma^c(\mathbf{w})$. At last, we will use the convergence of $\gamma^c(\mathbf{w}_t)$ to prove $\bar{\gamma}(\mathbf{w}_t)$ converges.

The following lemma shows that $\bar{\gamma}(\mathbf{w}) \approx \gamma^a(\mathbf{w})$.

**Lemma A.9** (Smoothed margin). *For the smooth margin $\gamma^a(\mathbf{w}_t)$ defined in (8) and the normalized margin $\bar{\gamma}(\mathbf{w}_t)$ defined in (3), we have*

- *When $L_t \leq \frac{1}{2n}$, we have*

$$q_{\min}(t) \leq -\log L_t \leq \log(2n) + q_{\min}(t),$$

  *and*

$$\bar{\gamma}(\mathbf{w}_t) \leq \gamma^a(\mathbf{w}_t) \leq \bar{\gamma}(\mathbf{w}_t) + \frac{\log(2n)}{\rho_t}.$$

- *Assume Assumption 1C holds. If $L_t \to 0$, then $|\gamma^a(\mathbf{w}_t) - \bar{\gamma}(\mathbf{w}_t)| \to 0$.*

*Proof of Lemma A.9.* To prove the first claim, notice that

$$L_t \leq \frac{1}{2n} \implies \ell(q_{\min}(t)) = \log(1 + \exp(-q_{\min}(t))) \leq nL_t \leq \frac{1}{2}.$$

Therefore we have

$$e^{-q_{\min}(t)} \leq e^{\frac{1}{2}} - 1 \leq 1.$$

Using $\frac{x}{2} \leq \log(1 + x) \leq x$ for $0 \leq x \leq 1$, we get

$$\frac{1}{2}e^{-q_{\min}(t)} \leq \ell(q_{\min}(t)) = \log(1 + e^{-q_{\min}(t)}) \leq e^{-q_{\min}(t)}.$$

Then we can bound $L_t$ by

$$\frac{1}{2n}e^{-q_{\min}(t)} \leq \frac{1}{n}\ell(q_{\min}(t)) \leq L_t \leq \ell(q_{\min}(t)) \leq e^{-q_{\min}(t)},$$

which is equivalent to

$$q_{\min}(t) \leq -\log L_t \leq \log(2n) + q_{\min}(t).$$

Dividing both sides by $\rho_t$ proves the second claim:

$$\bar{\gamma}(\mathbf{w}_t) := \frac{q_{\min}(t)}{\rho_t} \leq \gamma^a(\mathbf{w}_t) := \frac{-\log L_t}{\rho_t} \leq \bar{\gamma}(\mathbf{w}_t) + \frac{\log(2n)}{\rho_t} = \frac{\log(2n) + q_{\min}(t)}{\rho_t}.$$

For the last claim, we only need to show that $\rho_t \to \infty$. This is because if $L_t \to 0$, we have for any $i \in [n]$, $y_i f(\mathbf{w}_t; x_i) \to \infty$. Using $y_i f(\mathbf{w}_t; x_i) \leq C_{r,\kappa}\|\mathbf{w}_t\| + C_r$ from Lemma G.2, we have $\rho_t = \|\mathbf{w}_t\|_2 \to \infty$. □

The following lemma shows that $\gamma^c(\mathbf{w}) \approx \bar{\gamma}(\mathbf{w})$.

**Lemma A.10** (Modified and auxiliary margins). *Suppose that Assumption 1 holds. For the modified margin $\gamma^c(\mathbf{w}_t)$ defined in (10) and the auxiliary margin $\gamma^b(\mathbf{w}_t)$ defined in (9), we have*

- *If $L_t \leq \frac{1}{2ne^{\kappa+2}}$, there exists a constant $c$ such that*

$$\gamma^c(\mathbf{w}_t) \leq \gamma^b(\mathbf{w}_t) \leq \bar{\gamma}(\mathbf{w}_t) \leq \left(1 + \frac{c}{\log(1/L(\mathbf{w}_t))}\right)\gamma^c(\mathbf{w}_t).$$

*Proof of Lemma A.10.* To prove the first two inequalities, notice that

$$e^{\Phi(L_t)} = -\log(2ne^\kappa L_t) \cdot \exp\left(\frac{1 + (4\rho^2 + 2\beta)\tilde{\eta}}{\log(2ne^\kappa L_t)}\right) \qquad \text{using (10)}$$

$$\leq -\log(2ne^\kappa L_t). \quad \text{since } L_t \leq \frac{1}{2ne^\kappa}, \exp\left(\frac{1 + (4\rho^2 + 2\beta)\tilde{\eta}}{\log(2ne^\kappa L_t)}\right) \leq 1, \text{ and } \log(2ne^\kappa L_t) > 0$$

$$\leq -\log(L_t) - \log(2n)$$

$$\leq q_{\min}(t). \qquad\qquad \text{By argument 1 in Lemma A.9}$$

Using the above, (8) to (10), we have

$$\gamma^c(\mathbf{w}_t) := \frac{e^{\Phi(L(\mathbf{w}_t))}}{\|\mathbf{w}_t\|} \leq \frac{-\log(2ne^\kappa L_t)}{\|\mathbf{w}_t\|} =: \gamma^b(\mathbf{w}_t) \leq \frac{q_{\min}(t)}{\|\mathbf{w}_t\|} =: \bar{\gamma}(\mathbf{w}_t).$$

Then, we will prove the remaining inequality. First, we have

$$\frac{\bar{\gamma}(\mathbf{w}_t)}{\gamma^c(\mathbf{w}_t)} = \frac{\bar{\gamma}(\mathbf{w}_t)}{\gamma^b(\mathbf{w}_t)} \cdot \frac{\gamma^b(\mathbf{w}_t)}{\gamma^c(\mathbf{w}_t)}$$

$$= \frac{q_{\min}(t)}{-\log(2ne^\kappa L_t)} \cdot \exp\left(\frac{1 + (4\rho^2 + 2\beta)\tilde{\eta}}{-\log(2ne^\kappa L_t)}\right) \qquad \text{By the definitions of } \bar{\gamma}, \gamma^b, \gamma^c$$

$$\leq \frac{-\log(L_t)}{-\log(2ne^\kappa L_t)} \cdot \exp\left(\frac{1 + (4\rho^2 + 2\beta)\tilde{\eta}}{-\log(2ne^\kappa L_t)}\right) \qquad \text{Since } q_{\min}(t) \leq \log(-L_t) \text{ by Lemma A.9}$$

$$= \left(1 + \frac{\log(2ne^\kappa)}{-\log(2ne^\kappa L_t)}\right) \cdot \exp\left(\frac{1 + (4\rho^2 + 2\beta)\tilde{\eta}}{-\log(2ne^\kappa L_t)}\right).$$

To simplify the notation, we let $c_1 := 1 + (4\rho^2 + 2\beta)\tilde{\eta}$ and $c_2 = \log(2ne^\kappa)$. Since $L_t \leq \frac{1}{2ne^{\kappa+2}} \Rightarrow -\log(2ne^\kappa L_t) \geq 2 > 1$, we have

$$\frac{1 + (4\rho^2 + 2\beta)\tilde{\eta}}{-\log(2ne^\kappa L_t)} = \frac{c_1}{-\log(2ne^\kappa L_t)} \leq c_1.$$

Besides, given $x < c$, we have $e^x \leq 1 + e^c x$. Therefore,

$$\exp\left(\frac{1 + (4\rho^2 + 2\beta)\tilde{\eta}}{-\log(2ne^\kappa L_t)}\right) = \exp\left(\frac{c_1}{-\log(2ne^\kappa L_t)}\right) \leq 1 + \frac{c_1 \exp(c_1)}{-\log(2ne^\kappa L_t)}.$$

Plugging this into the bound for $\bar{\gamma}(\mathbf{w}_t)/\gamma^c(\mathbf{w}_t)$, we get

$$\frac{\bar{\gamma}(\mathbf{w}_t)}{\gamma^c(\mathbf{w}_t)} = \left(1 + \frac{c_2}{-\log(2ne^\kappa L_t)}\right) \cdot \exp\left(\frac{c_1}{-\log(2ne^\kappa L_t)}\right)$$

$$\leq \left(1 + \frac{c_2}{-\log(2ne^\kappa L_t)}\right) \cdot \left(1 + \frac{\exp(c_1)c_1}{-\log(2ne^\kappa L_t)}\right)$$

$$\leq 1 + \frac{c_2 + \exp(c_1)c_1 + c_2 c_1 \exp(c_1)}{-\log(2ne^\kappa L_t)} \qquad \text{Since } -\log(2ne^\kappa L_t) \geq 1$$

$$= 1 + \frac{c_2 + \exp(c_1)c_1 + c_2 c_1 \exp(c_1)}{-\log L_t - c_2}.$$

Note that $-\log L_t - c_2 \geq 2 > 1$. Because $\frac{x}{x-c_2}$ is decreasing when $x \geq c_2 + 1$, we have

$$\frac{-\log L_t}{-\log L_t - c_2} \leq c_2 + 1 \implies \frac{1}{-\log L_t - c_2} \leq \frac{c_2 + 1}{-\log L_t}.$$

Plug this inequality into the previous bound for $\bar{\gamma}(\mathbf{w}_t)/\gamma^c(\mathbf{w}_t)$ and we get

$$\frac{\bar{\gamma}(\mathbf{w}_t)}{\gamma^c(\mathbf{w}_t)} \leq 1 + \frac{(c_2 + \exp(c_1)c_1 + c_2 c_1 \exp(c_1))(c_2 + 1)}{-\log L_t}.$$

Let $c := (c_2 + \exp(c_1)c_1 + c_2 c_1 \exp(c_1))(c_2 + 1)$. We complete the proof. $\qquad \square$

The next lemma shows the convexity of $\Phi$ defined in (10). The convexity will help us analyze the change of $\gamma^c(\mathbf{w}_t)$ in the gradient descent dynamics. Specifically, we are going to use the property that

$$\Phi(x) - \Phi(y) \geq \Phi'(y)(x - y), \quad \text{for all } x, y.$$

**Lemma A.11** (Convexity of $\Phi$). *The function $\Phi(x)$ defined in (10) is convex for $0 < x < \frac{1}{2ne^{2+\kappa}}$.*

*Proof of Lemma A.11.* Check that

$$\Phi'(x) = \frac{1 - \frac{1}{\log(2ne^\kappa x)}(1 + (4\rho^2 + 2\beta)\tilde{\eta})}{x \log(2ne^\kappa x)},$$

and that

$$\Phi''(x) = \frac{(1 + (4\rho^2 + 2\beta)\tilde{\eta}) \cdot (2 + \log(2ne^\kappa x)) - \log^2(2ne^\kappa x) - \log(2ne^\kappa x)}{x^2 \log^3(2ne^\kappa x)}.$$

Note that when $x \leq \frac{1}{2ne^{2+\kappa}}$, we have $\log(2ne^\kappa x) \leq -2$, which implies

$$2 + \log(2ne^\kappa x) \leq 0, \ \log(2ne^\kappa x) < 0, \ \text{and} \ -\log^2(2ne^\kappa x) - \log(2ne^\kappa x) < 0.$$

Plugging these into the previous equality, we have $\Phi''(x) \geq 0$ when $0 < x < \frac{1}{2ne^{2+\kappa}}$. $\qquad \square$

Before we dive into the proof of the monotonic increasing $\gamma^c(\mathbf{w}_t)$, we show that $\gamma^c$ is bounded first. The convergence of $\gamma^c$ is a direct consequence of the monotonic increasing and the boundedness of $\gamma^c$.

**Lemma A.12** (An upper bound on $\gamma^c$, $\gamma^b$, $\gamma^a$ and $\gamma^c$)**.** *When* $L_t \leq \left\{ \frac{1}{2ne^\kappa}, \frac{1}{\tilde{\eta}(4\rho^2 + 2\beta)} \right\}$ *for* $t \geq s$, *there exists* $B_0$ *such that*

$$\gamma^c(\mathbf{w}_t) \leq \gamma^b(\mathbf{w}_t) \leq \gamma^a(\mathbf{w}_t) \leq \bar{\gamma}(\mathbf{w}_t) + \frac{\log 2n}{\rho_s} \leq B_0.$$

*Proof.* Apply lemma A.8, we have $\|\mathbf{w}_t\| \geq \rho_t \geq \rho_s$. Then we can apply lemma G.2 and there exists a constant $C_{\rho_s, \kappa}$ such that for all $i$,

$$|y_i f(\mathbf{w}_t; \mathbf{x}_i)| \leq C_{\rho_s, \kappa} \|\mathbf{w}_t\|.$$

Hence,

$$\bar{\gamma}(\mathbf{w}_t) = \frac{\arg\min_{i \in [n]} y_i f(\mathbf{w}_t; \mathbf{x}_i)}{\|\mathbf{w}_t\|} \leq C_{\rho_s, \kappa}.$$

Besides, by Lemma A.9, we have

$$\gamma^a(\mathbf{w}_t) \leq \bar{\gamma}(\mathbf{w}_t) + \frac{\log 2n}{\rho_t} \leq C_{\rho_s, \kappa} + \frac{\log 2n}{\rho_s}.$$

By Lemma A.10, we have

$$\gamma^c(\mathbf{w}_t) \leq \gamma^b(\mathbf{w}_t) \leq \gamma^a(\mathbf{w}_t) \leq C_{\rho_s, \kappa} + \frac{\log 2n}{\rho_s}.$$

Let $B_0 = C_{\rho_s, \kappa} + \frac{\log 2n}{\rho_s}$. Then, we complete the proof. $\qquad\square$

The following lemma is a variant of Proposition 5, item 1, in [Wu et al., 2024]. Before the lemma, we need some auxiliary definitions. let us define

$$\boldsymbol{\theta}_t := \frac{\mathbf{w}_t}{\|\mathbf{w}_t\|}, \quad \boldsymbol{\nu}_t := \boldsymbol{\theta}_t \boldsymbol{\theta}_t^\top (-\nabla L_t), \quad \boldsymbol{\mu}_t := \left(\mathbf{I} - \boldsymbol{\theta}_t \boldsymbol{\theta}_t^\top\right)(-\nabla L_t).$$

Therefore, we have

$$\|\nabla L_t\|^2 = \|\boldsymbol{\nu}_t\|^2 + \|\boldsymbol{\mu}_t\|^2.$$

The key point of this decomposition is that we consider the gradient of the loss function as a sum of two orthogonal components. The first component $\boldsymbol{\nu}_t$ is the component in the direction of $\mathbf{w}_t$, and the second component $\boldsymbol{\mu}_t$ is the component orthogonal to $\mathbf{w}_t$. We will show that the modified margin $\gamma^c(\mathbf{w}_t)$ is monotonically increasing. And the increase of $\gamma^c(\mathbf{w}_t)$ is lower bounded by a term that depends on $\|\boldsymbol{\mu}_t\|^2$.

**Lemma A.13** (Modified margin is monotonically increasing)**.** *Suppose Assumption 1 holds. If there exists $s$ such that*

$$L_s \leq \min\left\{ \frac{1}{e^{\kappa+2}2n}, \frac{1}{\tilde{\eta}(4\rho^2 + 2\beta)} \right\},$$

*then for $t \geq s$, we have*

- $L_{t+1} \leq L_t$.

- $v_t \geq -L_t \log(2ne^\kappa L_t) \geq 0$.

- $\rho_{t+1}^2 - \rho_t^2 \geq 2\tilde{\eta} v_t$.

- $\log \gamma^c(\mathbf{w}_{t+1}) - \log \gamma^c(\mathbf{w}_t) \geq \frac{\rho_t^2}{v_t^2} \|\boldsymbol{\mu}_t\|^2 \log \frac{\rho_{t+1}}{\rho_t}$.

*As a consequence, $\gamma^c(\mathbf{w}_t)$ admits a finite limit.*

*Proof of Lemma A.13.* The first claim is by Lemma A.3 and induction. The second and the third claims are consequences of Lemmas A.7 and A.8, respectively. We now prove the last claim. By Lemma A.3, we have

$$L_{t+1} - L_t := L(\mathbf{w}_{t+1}) - L(\mathbf{w}_t) \leq -\tilde{\eta}(1 - (2\rho^2 + \beta)\tilde{\eta}L_t)\|\nabla L_t\|^2.$$

Multiplying both sides by $2v_t \frac{1 - \frac{1}{2\log(2ne^\kappa L_t)}}{1 - (2\rho^2 + \beta)\tilde{\eta}L_t} > 0$, we get

$$\frac{1 - \frac{1}{2\log(2ne^\kappa L_t)}}{1 - (2\rho^2 + \beta)\tilde{\eta}L_t} 2v_t(L_{t+1} - L_t) \leq -2\tilde{\eta}v_t\left(1 - \frac{1}{2\log(2ne^\kappa L_t)}\right)\|\nabla L_t\|^2.$$

From Lemma A.8 we have

$$0 \leq \rho_{t+1}^2 - \rho_t^2 \leq 2\tilde{\eta}v_t\left(1 - \frac{1}{2\log(2ne^\kappa L_t)}\right).$$

Using the above we get

$$\frac{1 - \frac{1}{2\log(2ne^\kappa L_t)}}{1 - (2\rho^2 + \beta)\tilde{\eta}L_t} 2v_t(L_{t+1} - L_t) \leq -(\rho_{t+1}^2 - \rho_t^2)\|\nabla L_t\|^2. \tag{11}$$

Recall that

$$\|\nabla L_t\|^2 = \|\boldsymbol{\nu}_t\|^2 + \|\boldsymbol{\mu}_t\|^2.$$

For $\boldsymbol{\nu}_t$, we have

$$\|\boldsymbol{\nu}_t\| = \frac{1}{\rho_t}\langle \mathbf{w}_t, -\nabla L_t\rangle = \frac{v_t}{\rho_t}.$$

Then we can decompose $\|\nabla L_t\|^2$ as

$$\|\nabla L_t\|^2 = \|\boldsymbol{\nu}_t\|^2 + \|\boldsymbol{\mu}_t\|^2 = \frac{v_t^2}{\rho_t^2} + \|\boldsymbol{\mu}_t\|^2. \tag{12}$$

Plugging this into (11) and dividing both two sides by $2v_t^2$, we have

$$\frac{1 - \frac{1}{2\log(2ne^\kappa L_t)}}{(1 - (2\rho^2 + \beta)\tilde{\eta}L_t)v_t}(L_{t+1} - L_t) \leq -\frac{1}{\rho_t^2}(\rho_{t+1}^2 - \rho_t^2)\left(\frac{1}{2} + \frac{\rho_t^2}{2v_t^2}\|\boldsymbol{\mu}_t\|^2\right).$$

By Lemma A.7, we have $v_t \geq -L_t\log(2ne^\kappa L_t)$. Define

$$\Psi(x) := -\frac{1 - \frac{1}{2\log(2ne^\kappa x)}}{(1 - (2\rho^2 + \beta)\tilde{\eta}x)x\log(2ne^\kappa x)}.$$

Then we have

$$\Psi(L_t)(L_{t+1} - L_t) := -\frac{1 - \frac{1}{2\log(2ne^\kappa L_t)}}{(1 - (2\rho^2 + \beta)\tilde{\eta}L_t)L_t\log(2ne^\kappa L_t)}(L_{t+1} - L_t)$$

$$\leq \frac{1 - \frac{1}{2\log(2ne^\kappa L_t)}}{(1 - (2\rho^2 + \beta)\tilde{\eta}L_t)v_t}(L_{t+1} - L_t) \tag{13}$$

$$\leq -\frac{1}{\rho_t^2}(\rho_t^2 - \rho_{t+1}^2)\left(\frac{1}{2} + \frac{\rho_t^2}{2v_t^2}\|\boldsymbol{\mu}_t\|^2\right).$$

We are going to show that $\Psi(x) \leq -\Phi'(x)$. Note that when $0 < x \leq \min\left\{\frac{1}{e^{\kappa+2}2n}, \frac{1}{\tilde{\eta}(4\rho^2+2\beta)}\right\}$, we have $\log(2ne^\kappa x) < 0$ and $1 - (2\rho^2 + \beta)\tilde{\eta}x \geq \frac{1}{2} > 0$. Therefore, we have

$$\Psi(x) = \underbrace{\frac{1 - \frac{1}{2\log(2ne^\kappa x)}}{1 - (2\rho^2 + \beta)\tilde{\eta}x}}_{=:J>0} \cdot \underbrace{\frac{-1}{x\log(2ne^\kappa x)}}_{>0}. \tag{14}$$

To get an upper bound of $\Psi(x)$, we just need an upper bound of $J$. Let $a := \frac{-1}{2\log(2ne^\kappa x)} > 0$ and $b := (2\rho^2 + \beta)\tilde{\eta}x \in (0, 1/2]$. Then we invoke Lemma G.4 to get

$$J := \frac{1 + a}{1 - b} \leq 1 + 2a + 2b = 1 - \frac{1}{\log(2ne^\kappa x)} + (4\rho^2 + 2\beta)\tilde{\eta}x.$$

Recall that $x \leq \frac{1}{e^{\kappa+2}2n} \leq \frac{1}{2ne^\kappa}$ and $2ne^\kappa \geq 1$. Then we apply Lemma G.3 to get $x \leq \frac{-1}{\log(2ne^\kappa x)}$. Plugging this into the bound of $J$, we get

$$J \leq 1 - \frac{1}{\log(2ne^\kappa x)} + (4\rho^2 + 2\beta)\tilde{\eta}x \leq 1 - \frac{1}{\log(2ne^\kappa x)}(1 + (4\rho^2 + 2\beta)\tilde{\eta}). \tag{15}$$

Plugging (15) into (14), we have

$$\Psi(x) = J \cdot \frac{-1}{x\log(2ne^\kappa x)} \leq -\frac{1 - \frac{1}{\log(2ne^\kappa x)}(1 + (4\rho^2 + 2\beta)\tilde{\eta})}{x\log(2ne^\kappa x)} = -\Phi'(x),$$

which verifies that $\Psi(x) \leq -\Phi'(x)$. By this and (13), we have

$$\Phi'(L_t)(L_{t+1} - L_t) + \varphi'(\rho_t^2)(\rho_{t+1}^2 - \rho_t^2)\left(\frac{1}{2} + \frac{\rho_t^2}{2v_t^2}\|\boldsymbol{\mu}_t\|^2\right) \geq 0,$$

where $\varphi(x) = -\log x = \log(1/x)$. Recall that for $0 < x \leq \frac{1}{2ne^{\kappa+2}}$, $\Phi(x)$ is convex by Lemma A.11. By convexity of $\varphi$ and $\Phi$, we have

$$\Phi(L_{t+1}) - \Phi(L_t) + \left(\log\frac{1}{\rho_{t+1}^2} - \log\frac{1}{\rho_t^2}\right)\left(\frac{1}{2} + \frac{\rho_t^2}{2v_t^2}\|\boldsymbol{\mu}_t\|^2\right) \geq 0.$$

By the definition of $\gamma^c$ in (10), this can be rewritten as

$$\log\gamma^c(\mathbf{w}_{t+1}) - \log\gamma^c(\mathbf{w}_t) = (\Phi(L_{t+1}) - \Phi(L_t)) + \left(\log\frac{1}{\rho_{t+1}} - \log\frac{1}{\rho_t}\right)$$
$$\geq -\left(\log\frac{1}{\rho_{t+1}^2} - \log\frac{1}{\rho_t^2}\right)\frac{\rho_t^2}{2v_t^2}\|\boldsymbol{\mu}_t\|^2$$
$$= \frac{\rho_t^2}{v_t^2}\|\boldsymbol{\mu}_t\|^2\log\frac{\rho_{t+1}}{\rho_t}$$
$$\geq 0,$$

where the last inequality is because of Lemma A.8. We have shown that $\gamma^c(\mathbf{w}_t)$ is monotonically increasing. By Lemma A.12, $\gamma^c(\mathbf{w}_t)$ is bounded. Therefore $\gamma^c(\mathbf{w}_t)$ admits a finite limit. This completes the proof. $\qquad\square$

### A.4 Sharp rates of Loss and Parameter Norm

Right now, we have already proved that $\gamma^c(\mathbf{w}_t)$ is monotonically increasing and bounded, which indicates $\gamma^c(\mathbf{w}_t)$ converges. However, if we want to show that $\bar{\gamma}(\mathbf{w}_t)$ converges, we still need to verify that $L_t \to 0$, which is the crucial condition for $\gamma^c(\mathbf{w}_t), \gamma^b(\mathbf{w}_t), \gamma^a(\mathbf{w}_t)$, and $\bar{\gamma}(\mathbf{w}_t)$ to share the same limit, by Lemma A.9 and Lemma A.10.

Fortunately, with the monotonicity of $\gamma^c(\mathbf{w}_t)$, we can prove that $L_t$ converges to zero and even characterize the rate of $L_t$.

**Lemma A.14** (Rate of $L_t$ in general model). *Suppose Assumption 1 holds. If there is an $s$ such that*

$$L(\mathbf{w}_s) \leq \min\left\{\frac{1}{e^{\kappa+2}2n}, \frac{1}{\tilde{\eta}(4\rho^2 + 2\beta)}\right\},$$

*then for every $t \geq s$ we have*

$$\frac{1}{\frac{1}{L(\mathbf{w}_s)} + 3\tilde{\eta}\rho^2(t-s)} \leq L(\mathbf{w}_t) \leq \frac{2}{(t-s)\tilde{\eta}\gamma^c(\mathbf{w}_s)^2}.$$

*That is, $L(\mathbf{w}_t) = \Theta(1/t) \to 0$ as $t \to \infty$.*

*Proof of Lemma A.14.* By Lemma A.3 and (12) in the proof of Lemma A.13, we know $L_t$ is decreasing and

$$L_{t+1} - L_t \leq -\frac{\tilde{\eta}}{2}\|\nabla L_t\|^2 \leq -\frac{\tilde{\eta}}{2}\|\boldsymbol{\nu}_t\|_2^2 \leq -\frac{\tilde{\eta}}{2}\frac{v_t^2}{\rho_t^2}. \tag{16}$$

We will establish an upper bound for $\rho_t$ first. Note that $\gamma^c(\mathbf{w}_t)$ is increasing for $t \geq s$ by Lemma A.13 and $\gamma^b(\mathbf{w}_t) \geq \gamma^c(\mathbf{w}_t)$ by Lemma A.10. By (9), we have

$$\rho_t = \frac{-\log(2ne^\kappa L_t)}{\gamma^b(\mathbf{w}_t)} \leq \frac{-\log(2ne^\kappa L_t)}{\gamma^c(\mathbf{w}_t)} \leq \frac{-\log(2ne^\kappa L_t)}{\gamma^c(\mathbf{w}_s)}.$$

Combining this with Lemma A.7, we have

$$\frac{v_t}{\rho_t} \geq \frac{-L_t \log(2ne^\kappa L_t)}{\frac{-\log(2ne^\kappa L_t)}{\gamma^c(\mathbf{w}_s)}} = L_t \gamma^c(\mathbf{w}_s).$$

Plugging this into (16), we have

$$L_{t+1} - L_t \leq -\frac{\tilde{\eta}}{2} L_t^2 \gamma^c(\mathbf{w}_s)^2,$$

which implies

$$\frac{\tilde{\eta}\gamma(\mathbf{w}_s)^2}{2} \leq \frac{L_t - L_{t+1}}{L_t^2}$$

$$\leq \frac{L_t - L_{t+1}}{L_t L_{t+1}} \qquad\qquad \text{Since } L_{t+1} \leq L_t$$

$$= \frac{1}{L_{t+1}} - \frac{1}{L_t}, \quad t \geq s.$$

Telescoping the sum from $s$ to $t$, we have

$$(t-s)\frac{\tilde{\eta}\gamma^c(\mathbf{w}_s)^2}{2} \leq \frac{1}{L_t} - \frac{1}{L_s} \leq \frac{1}{L_t}.$$

Therefore we have

$$L_t \leq \frac{2}{(t-s)\tilde{\eta}\gamma^c(\mathbf{w}_s)^2}.$$

Next we show the lower bound on the risk. By Lemma A.3 we have

$$L_{t+1} - L_t \geq -\tilde{\eta}(1 + \beta\tilde{\eta}L_t)\|\nabla L_t\|^2 \geq -\frac{3}{2}\tilde{\eta}\|\nabla L_t\|^2.$$

Observe that under Assumption 1A,

$$\|\nabla L_t\| = \left\|\frac{1}{n}\sum_{i=1}^{n} \ell'(q_i(t))y_i \nabla f(\mathbf{w}_t; \mathbf{x}_i)\right\| \leq \rho L_t.$$

Then we have

$$L_{t+1} - L_t \geq -\tilde{\eta}\frac{3}{2}\rho^2 L_t^2, \quad t \geq s.$$

Let $\tilde{L}_t := \frac{3\tilde{\eta}\rho^2}{2}L_t$, we have $\tilde{L}_s \leq \frac{3\tilde{\eta}\rho^2}{2}\frac{1}{\tilde{\eta}(4\rho^2 + 2\beta)} \leq \frac{3}{8} \leq \frac{1}{2}$. Furthermore, since $L_t$ decreases monotonically, $\tilde{L}_t \leq \tilde{L}_s \leq \frac{1}{2}$. The inequality becomes

$$\tilde{L}_{t+1} - \tilde{L}_t \geq -\tilde{L}_t^2.$$

Therefore, let $c = \frac{1}{\tilde{L}_s}$ and apply Lemma G.1, we have for any $t \geq s$,

$$\tilde{L}_t \geq \frac{1}{c + 2(t-s)}.$$

This is equivalent to

$$L_t \geq \frac{1}{\frac{1}{L_s} + 3\tilde{\eta}\rho^2(t-s)}.$$

We have completed the proof. $\qquad\qquad\qquad\qquad\qquad\qquad\qquad\qquad\qquad\qquad\qquad\quad$ □

Furthermore, we can characterize the order of $\rho_t$ in the stable phase.

**Lemma A.15** (Order of $\rho_t$ in general model). *Suppose Assumption 1 holds. If there is $s$ such that*

$$L(\mathbf{w}_s) \leq \min\left\{\frac{1}{e^{\kappa+2}2n}, \ \frac{1}{\tilde{\eta}(4\rho^2+2\beta)}\right\},$$

*then for $t \geq s$ we have*

$$\rho_t = \Theta(\log(t)).$$

*Proof of Lemma A.15.* Note that $\gamma^c(\mathbf{w}_t)$ is increasing for $t \geq s$ by Lemma A.13 and $\gamma^b(\mathbf{w}_t) \geq \gamma^c(\mathbf{w}_t)$ by Lemma A.10. Therefore,

$$\rho_t \leq \frac{-\log(2ne^\kappa L_t)}{\gamma^b(\mathbf{w}_t)} \leq \frac{-\log(2ne^\kappa L_t)}{\gamma^c(\mathbf{w}_t)} \leq \frac{-\log(2ne^\kappa L_t)}{\gamma^c(\mathbf{w}_s)}.$$

Combining this with Lemma A.14, we have

$$\rho_t \leq \frac{\log\frac{1/L(\mathbf{w}_s)+3\tilde{\eta}\rho^2(t-s)}{2ne^\kappa}}{\gamma^c(\mathbf{w}_s)} = \mathcal{O}(\log(\tilde{\eta}t)).$$

Besides, we have $q_{\min} \geq \iota(\log\frac{1}{L} - \log n)$ by Lemma A.9 and $q_{\min} \leq B_0\rho_t$ by Lemma A.12. Therefore we have

$$\rho_t \geq \frac{\iota(\log\frac{1}{nL_t})}{B_0} \geq \frac{\log\frac{1}{nL_t}}{2B_0} \geq \frac{\log\frac{(t-s)\tilde{\eta}\gamma^c(\mathbf{w}_s)^2}{2n}}{2B_0} = \Omega(\log(t)),$$

where the second inequality is because for $\iota(x)$ defined in (6), $\iota(x) \geq \frac{x}{2}$ for $x \geq 0.6$, and the third inequality is by Lemma A.14. Combining them, we get

$$\rho_t = \Theta(\log(t)).$$

This completes the proof. □

## A.5 Proof of Theorem 2.2

*Proof of Theorem 2.2.* We prove the items one by one.

- The monotonicity of $L_t$ comes from the result of Lemma A.3 directly.

- Item 1 is due to Lemma A.14 .

- For item 2, the monotonicity of $\rho_t$ comes from the result of Lemma A.8 and the order is due to Lemma A.15.

- For item 3, we first know that $L_t \to 0$ by Lemma A.14. Then, by Lemma A.13 and Lemma A.12, we know that $\gamma^c(\mathbf{w}_t)$ converges. Combining these with Lemma A.9 and Lemma A.10, we know that $\gamma^c(\mathbf{w}_t)$ is an $\left(1 + O\left(1/\left(\log\frac{1}{L(\mathbf{w}_t)}\right)\right)\right)$-multiplicative approximation of $\bar{\gamma}(\mathbf{w}_t)$, and $\bar{\gamma}(\mathbf{w}_t)$ shares the same limit as $\gamma^c(\mathbf{w}_t)$.

□

## B EoS Phase Analysis

In this section, we focus on the linearly separable case, that is, we work under Assumption 3. We mainly follow the idea of [Wu et al., 2024] for the proof. In detail, we consider a comparator

$$\mathbf{u} := \mathbf{u}_1 + \mathbf{u}_2,$$

where where

$$\mathbf{u}_1 := \begin{pmatrix} \mathbf{u}_1^{(1)} \\ \vdots \\ \mathbf{u}_1^{(m)} \end{pmatrix}, \quad \text{with } \mathbf{u}_1^{(j)} := a_j \frac{\log(\gamma^2\eta T)+\kappa}{\alpha\gamma} \cdot \mathbf{w}_*, \quad j = 1,\ldots,m, \tag{17}$$

and

$$\mathbf{u}_2 := \begin{pmatrix} \mathbf{u}_2^{(1)} \\ \vdots \\ \mathbf{u}_2^{(m)} \end{pmatrix}, \quad \text{with} \ \ \mathbf{u}_2^{(j)} := a_j \frac{\eta}{2\gamma} \cdot \mathbf{w}_*, \quad j = 1, \dots, m. \tag{18}$$

Consider the following decomposition,

$$\|\mathbf{w}_{t+1} - \mathbf{u}\|^2 = \|\mathbf{w}_t - \mathbf{u}\|^2 + 2m\eta \langle \nabla L(\mathbf{w}_t), \mathbf{u} - \mathbf{w}_t \rangle + m^2 \eta^2 \|\nabla L(\mathbf{w}_t)\|^2$$
$$= \|\mathbf{w}_t - \mathbf{u}\|^2 + 2m\eta \underbrace{\langle \nabla L(\mathbf{w}_t), \mathbf{u}_1 - \mathbf{w}_t \rangle}_{=:I_1(\mathbf{w}_t)} + m\eta \Big( \underbrace{2\langle \nabla L(\mathbf{w}_t), \mathbf{u}_2 \rangle + m\eta \|\nabla L(\mathbf{w}_t)\|^2}_{=:I_2(\mathbf{w}_t)} \Big).$$

We aim to prove $I_1(\mathbf{w}_t) \leq \frac{1}{T} - L(\mathbf{w}_t)$ and $I_2(\mathbf{w}_t) \leq 0$. Then we can get a bound for the average loss by telescope summing the decomposition. Here we also introduced the following vector $\bar{\mathbf{w}}_*$:

$$\bar{\mathbf{w}}_* := \begin{pmatrix} a_1 \mathbf{w}_* \\ \vdots \\ a_n \mathbf{w}_* \end{pmatrix}$$

We can observe that $\mathbf{u}_1 = \frac{\log(\gamma^2 \eta T) + \kappa}{\alpha \gamma} \bar{\mathbf{w}}_*$ and $\mathbf{u}_2 = \frac{\eta}{2\gamma} \bar{\mathbf{w}}_*$.

**Lemma B.1** (A bound on $I_1(\mathbf{w})$ in the EoS phase). *For $\mathbf{u}_1$ defined in* (17), *we have*

$$I_1(\mathbf{w}) := \langle \nabla L(\mathbf{w}), \mathbf{u}_1 - \mathbf{w} \rangle \leq \frac{1}{\gamma^2 \eta T} - L(\mathbf{w}).$$

*Proof.* Since $L$ is averaged over the individual losses incurred at the data $(\mathbf{x}_i, y_i)_{i=1}^n$ and gradient is a linear operator, it suffices to prove the claim assuming there is only a single data point $(\mathbf{x}, y)$. Then by Assumption 3, we have

$$\langle y\mathbf{x}, \mathbf{w}_* \rangle \geq \gamma > 0.$$

Then the loss becomes

$$L(\mathbf{w}) = \ell(yf(\mathbf{w}; \mathbf{x})) = \ell\Big( y \frac{1}{m} \sum_{j=1}^m a_j \phi(\mathbf{x}^\top \mathbf{w}^{(j)}) \Big).$$

Now we expand $I_1(\mathbf{w})$:

$$I_1(\mathbf{w}) := \langle \nabla L(\mathbf{w}), \mathbf{u}_1 - \mathbf{w} \rangle$$
$$= \ell'\big(yf(\mathbf{w}; \mathbf{x})\big) \langle y \nabla f(\mathbf{w}; \mathbf{x}), \mathbf{u}_1 - \mathbf{w} \rangle$$
$$= \ell'\big(yf(\mathbf{w}; \mathbf{x})\big) \frac{1}{m} \sum_{k=1}^m a_k y \phi'(x^\top \mathbf{w}^{(k)}) \mathbf{x}^\top (\mathbf{u}_1^{(k)} - \mathbf{w}^{(k)})$$
$$= \ell'\big(yf(\mathbf{w}; \mathbf{x})\big) \Bigg[ \underbrace{\frac{1}{m} \sum_{k=1}^m a_k y \Big( \phi'(\mathbf{x}^\top \mathbf{w}^{(k)}) \mathbf{x}^\top \mathbf{u}_1^{(k)} + \phi(\mathbf{x}^\top \mathbf{w}^{(k)}) - \phi'(\mathbf{x}^\top \mathbf{w}^{(k)}) \mathbf{x}^\top \mathbf{w}^{(k)} \Big)}_{=:J_1}$$
$$\underbrace{- \frac{1}{m} \sum_{k=1}^m a_k y \phi(\mathbf{x}^\top \mathbf{w}^{(k)})}_{=:J_2} \Bigg]. \tag{19}$$

By definition we have $J_2 = yf(\mathbf{w}; \mathbf{x})$. As for $J_1$, using $\phi' \geq \alpha$ and $a_k y x^\top \mathbf{u}_1^{(k)} \geq 0$ by Assumption 3, we have

$$J_1 := \frac{1}{m} \sum_{k=1}^m a_k y \Big( \phi'(\mathbf{x}^\top \mathbf{w}^{(k)}) \mathbf{x}^\top \mathbf{u}_1^{(k)} + \phi(\mathbf{x}^\top \mathbf{w}^{(k)}) - \phi'(\mathbf{x}^\top \mathbf{w}^{(k)}) \mathbf{x}^\top \mathbf{w}^{(k)} \Big)$$
$$\geq \frac{1}{m} \sum_{k=1}^m a_k \alpha y \mathbf{x}^\top \mathbf{u}_1^{(k)} + \frac{1}{m} \sum_{k=1}^m a_k y \Big( \phi(\mathbf{x}^\top \mathbf{w}^{(k)}) - \phi'(\mathbf{x}^\top \mathbf{w}^{(k)}) \mathbf{x}^\top \mathbf{w}^{(k)} \Big)$$

$$\geq \frac{1}{m}\sum_{k=1}^{m} a_k^2 \alpha \frac{\log(\gamma^2 \eta T) + \kappa}{\alpha \gamma} y\mathbf{x}^\top \mathbf{w}_* - \frac{1}{m}\sum_{k=1}^{m} |a_k|\kappa$$

$$\text{since } |\phi(\mathbf{x}^\top \mathbf{w}^{(k)}) - \phi'(\mathbf{x}^\top \mathbf{w}^{(k)})\mathbf{x}^\top \mathbf{w}^{(k)}| \leq \kappa \text{ by Assumption 2C}$$

$$\geq \log(\gamma^2 \eta T) + \kappa - \kappa$$

$$\text{since } y\mathbf{x}^\top \mathbf{w}_* \geq \gamma \text{ and } \sum_{k=1}^{m} a_k^2 = m$$

$$= \log(\gamma^2 \eta T). \tag{20}$$

Plugging in $J_2 = yf(\mathbf{w}; \mathbf{x})$ and (20) into (19), we get

$$I_1(\mathbf{w}) = \langle \nabla L(\mathbf{w}), \mathbf{u}_1 - \mathbf{w} \rangle = \ell'\big(yf(\mathbf{w}; \mathbf{x})\big)(J_1 - J_2)$$

$$\leq \ell'\big(yf(\mathbf{w}; \mathbf{x})\big)\Big[\log(\gamma^2 \eta T) - yf(\mathbf{w}; \mathbf{x})\Big] \qquad \text{since } \ell' < 0$$

$$\leq \ell(\log(\gamma^2 \eta T)) - \ell(yf(\mathbf{w}; \mathbf{x})) \qquad \text{since } \ell \text{ is convex}$$

$$\leq \frac{1}{\gamma^2 \eta T} - L(\mathbf{w}).$$

where in the last inequality, we use $\ell(x) \leq \exp(-x)$ and we only consider a single data point. This completes the proof. $\qquad\square$

**Lemma B.2** (A bound on $I_2(\mathbf{w})$ in EoS). *For $\mathbf{u}_2$ defined in (18), for every $\mathbf{w}$,*

$$I_2(\mathbf{w}) := 2\langle \nabla L(\mathbf{w}), \mathbf{u}_2 \rangle + m\eta \|\nabla L(\mathbf{w})\|^2 \leq 0.$$

*Proof.* For simplicity, we define

$$g_i(\mathbf{w}^{(j)}) := \ell'(y_i f(\mathbf{w}; \mathbf{x}_i))\phi'(\mathbf{x}_i^\top \mathbf{w}^{(j)}).$$

Note that $-1 \leq \ell'(\cdot) \leq 0$ and $0 < \alpha \leq \phi'(\cdot) \leq 1$, we have

$$-1 \leq g_i(\mathbf{w}^{(j)}) \leq 0.$$

Under this notation, we have

$$\frac{\partial L(\mathbf{w})}{\partial \mathbf{w}_i} = \frac{1}{n}\sum_{i=1}^{n} \ell'\big(y_i f(\mathbf{w}; \mathbf{x}_i)\big) y_i a_j m^{-1} \phi'(\mathbf{x}_i^\top \mathbf{w}^{(j)})\mathbf{x}_i$$

$$= \frac{1}{n}\sum_{i=1}^{n} g_i(\mathbf{w}^{(j)}) a_j m^{-1} y_i \mathbf{x}_i.$$

So we have

$$I_2(\mathbf{w}) := 2\langle \nabla L(\mathbf{w}), \mathbf{u}_2 \rangle + m\eta \|\nabla L(\mathbf{w})\|^2$$

$$= \frac{1}{m}\sum_{j=1}^{m}\left[\frac{2}{n}\sum_{i=1}^{n} g_i(\mathbf{w}^{(j)}) a_j y_i \cdot \mathbf{x}_i^\top \mathbf{u}_2^{(j)} + \eta \left\|\frac{1}{n}\sum_{i=1}^{n} g_i(\mathbf{w}^{(j)}) a_j y_i \mathbf{x}_i\right\|^2\right].$$

For the term inside the bracket, we have

$$\frac{2}{n}\sum_{i=1}^{n} g_i(\mathbf{w}^{(j)}) a_j y_i \cdot \mathbf{x}_i^\top \mathbf{u}_2^{(j)} + \eta \left\|\frac{1}{n}\sum_{i=1}^{n} g_i(\mathbf{w}^{(j)}) a_j y_i \mathbf{x}_i\right\|^2$$

$$= \frac{2}{n}\sum_{i=1}^{n} g_i(\mathbf{w}^{(j)}) a_j y_i \cdot \mathbf{x}_i^\top \frac{\eta}{2\gamma} a_j \mathbf{w}_* + \eta \left\|\frac{1}{n}\sum_{i=1}^{n} g_i(\mathbf{w}^{(j)}) a_j y_i \mathbf{x}_i\right\|^2 \qquad \text{since } \mathbf{u}_2^{(j)} := \frac{\eta a_j}{2\gamma}\mathbf{w}_* \text{ by (18)}$$

$$\leq \frac{2}{n}\sum_{i=1}^{n} g_i(\mathbf{w}^{(j)}) a_j^2 \frac{\eta}{2\gamma}\gamma + \eta \left\|\frac{1}{n}\sum_{i=1}^{n} g_i(\mathbf{w}^{(j)}) a_j y_i \mathbf{x}_i\right\|^2 \qquad \text{since } g_i(\cdot) \leq 0 \text{ and } y_i \mathbf{x}_i^\top \mathbf{w}_* \geq \gamma$$

$$= \eta\left(\frac{1}{n}\sum_{i=1}^{n} g_i(\mathbf{w}^{(j)}) + \left\|\frac{1}{n}\sum_{i=1}^{n} g_i(\mathbf{w}^{(j)}) y_i \mathbf{x}_i\right\|^2\right) \qquad \text{since } a_j^2 = 1$$

$$\leq \eta\left(\frac{1}{n}\sum_{i=1}^{n}g_i(\mathbf{w}^{(j)}) + \frac{1}{n}\sum_{i=1}^{n}g_i^2(\mathbf{w}^{(j)})\right) \qquad \text{since } |g_i(\cdot)| \leq 1 \text{ and } \|y\mathbf{x}\| \leq 1$$

$$\leq 0. \qquad \text{since } -1 \leq g_i(\cdot) \leq 0$$

Hence, we prove that $I_2(\mathbf{w}) \leq 0$. $\qquad\square$

**Theorem B.3** (A split optimization bound). *For every $\eta > 0$ and $\mathbf{u} = \mathbf{u}_1 + \mathbf{u}_2$ such that*

$$\mathbf{u}_1 := \begin{pmatrix} \mathbf{u}_1^{(1)} \\ \vdots \\ \mathbf{u}_1^{(m)} \end{pmatrix}, \quad \text{with } \mathbf{u}_1^{(j)} := a_j \frac{\log(\gamma^2\eta t) + \kappa}{\alpha\gamma} \cdot \mathbf{w}_*, \quad j = 1,\dots,m,$$

*and*

$$\mathbf{u}_2 := \begin{pmatrix} \mathbf{u}_2^{(1)} \\ \vdots \\ \mathbf{u}_2^{(m)} \end{pmatrix}, \quad \text{with } \mathbf{u}_2^{(j)} := a_j \frac{\eta}{2\gamma} \cdot \mathbf{w}_*, \quad j = 1,\dots,m.$$

*we have:*

$$\frac{\|\mathbf{w}_T - \mathbf{u}\|^2}{2m\eta T} + \frac{1}{T}\sum_{k=0}^{T-1} L(\mathbf{w}^{(k)}) \leq \frac{1 + 8\log^2(\gamma^2\eta T)/\alpha^2 + 8\kappa^2/\alpha^2 + \eta^2}{\gamma^2\eta T} + \frac{\|\mathbf{w}_0\|^2}{m\eta T},$$

*for all $T$.*

*Proof.* By Lemma B.1 and Lemma B.2, we have

$$\|\mathbf{w}_{t+1} - \mathbf{u}\|^2 = \|\mathbf{w}_t - \mathbf{u}\|^2 + 2m\eta I_1(\mathbf{w}_t) + \eta m I_2(\mathbf{w}_t)$$
$$\leq \|\mathbf{w}_t - \mathbf{u}\|^2 + 2m\eta I_1(\mathbf{w}_t)$$
$$\leq \|\mathbf{w}_t - \mathbf{u}\|^2 + 2m\eta\left(\frac{1}{\gamma^2\eta T} - L(\mathbf{w}_t)\right).$$

Telescoping the sum, we get

$$\frac{\|\mathbf{w}_T - \mathbf{u}\|^2}{2m\eta} + \sum_{t=0}^{T-1} L(\mathbf{w}_t) \leq 1 + \frac{\|\mathbf{w}_0 - \mathbf{u}\|^2}{2m\eta}.$$

By (17) and (18), we have

$$\|\mathbf{w}_0 - \mathbf{u}\|^2 \leq 2\|\mathbf{w}_0\|^2 + 2\|\mathbf{u}\|^2$$
$$\leq 2\|\mathbf{w}_0\|_2^2 + 4\|\mathbf{u}_1\|^2 + 4\|\mathbf{u}_2\|^2$$
$$= 2\|\mathbf{w}_0\|_2^2 + \frac{8m\log(\gamma^2\eta T)^2 + 8m\kappa^2}{\alpha^2\gamma^2} + \frac{m\eta^2}{\gamma^2},$$

which implies that

$$\frac{\|\mathbf{w}_T - \mathbf{u}\|^2}{2m\eta t} + \frac{1}{T}\sum_{k=0}^{T-1} L(\mathbf{w}^{(k)}) \leq \frac{1 + 8\log^2(\gamma^2\eta T)/\alpha^2 + 8\kappa^2/\alpha^2 + \eta^2}{\gamma^2\eta T} + \frac{\|\mathbf{w}_0\|^2}{m\eta T}.$$

We complete the proof. $\qquad\square$

## B.1 Proof of Theorem 3.2

*Proof of Theorem 3.2.* By Theorem B.3, we have

$$\frac{1}{T}\sum_{k=0}^{T-1} L(\mathbf{w}^{(k)}) \leq \frac{1 + 8\log^2(\gamma^2\eta T)/\alpha^2 + 8\kappa^2/\alpha^2 + \eta^2}{\gamma^2\eta T} + \frac{\|\mathbf{w}_0\|^2}{m\eta T}.$$

This completes the proof. $\qquad\square$

## C  Phase Transition Analysis

In this section, we will analyze the phase transition. In detail, we follow the idea of [Wu et al., 2024] and apply the perceptron argument [Novikoff, 1962] to locate the phase transition time. Compare to the previous EoS phase analysis, we need an extra assumption on the smoothness of the activation function, which is the Assumption 2B.

To proceed, let us define the following quantities for the GD process:

$$G(\mathbf{w}) := \frac{1}{n} \sum_{i=1}^{n} \frac{1}{1 + \exp\left(y_i f(\mathbf{w}; \mathbf{x}_i)\right)}, \quad F(\mathbf{w}) := \frac{1}{n} \sum_{i=1}^{n} \exp\left(-y_i f(\mathbf{w}; \mathbf{x}_i)\right).$$

Due to the self-boundedness of the logistic function, we can show that $G(\mathbf{w}), L(\mathbf{w}), F(\mathbf{w})$ are equivalent in the following sense.

**Lemma C.1** (Equivalence of $G, L, F$)**.**

    *1.* $G(\mathbf{w}) \leq L(\mathbf{w}) \leq F(\mathbf{w})$.

    *2.* $\alpha\gamma G(\mathbf{w}) \leq \sqrt{m}\|\nabla L(\mathbf{w})\| \leq G(\mathbf{w})$.

    *3.* *If* $G(\mathbf{w}) \leq \frac{1}{2n}$, *then* $F(\mathbf{w}) \leq 2G(\mathbf{w})$.

*Proof.* The first claim is by the property of the logistic loss. For the second one,

$$
\begin{aligned}
\|\nabla L(\mathbf{w})\|^2 &= \sum_{j=1}^{m} \left\| \frac{1}{n} \sum_{i=1}^{n} \ell'(y_i f(\mathbf{w}; \mathbf{x}_i)) \cdot y_i \cdot a_j m^{-1} \phi(\mathbf{x}_i^\top \mathbf{w}^{(j)}) \mathbf{x}_i \right\|_2^2 \\
&\leq \sum_{j=1}^{m} \left( \frac{1}{n} \sum_{i=1}^{n} \ell'(y_i f(\mathbf{w}; \mathbf{x}_i)) \cdot m^{-1} \right)^2 \qquad\qquad \text{since } \|y_i a_j \phi(\mathbf{x}_i^\top \mathbf{w}^{(j)}) \mathbf{x}_i\| \leq 1 \\
&= \frac{1}{m} G^2(\mathbf{w}).
\end{aligned}
$$

Besides, we have

$$
\begin{aligned}
\sqrt{m}\|\nabla L(\mathbf{w})\| &\geq \langle -\nabla L(\mathbf{w}), \bar{\mathbf{w}}_* \rangle \qquad\qquad\qquad\qquad\qquad\quad \text{since } \|\bar{\mathbf{w}}_*\| \leq \sqrt{m} \\
&= -\frac{1}{nm} \sum_{i=1}^{n} \sum_{j=1}^{m} \ell'(y_i f(\mathbf{w}; \mathbf{x}_i)) y_i \phi'(\mathbf{x}_i^\top \mathbf{w}_*) \mathbf{x}_i^\top \mathbf{w}_* \\
&\geq \alpha\gamma \frac{1}{n} \sum_{i=1}^{n} \frac{1}{1 + \exp\left(y_i f(\mathbf{w}; \mathbf{x}_i)\right)} \qquad\quad \text{since } \phi' \geq \alpha \text{ and } y_i \mathbf{x}_i^\top \mathbf{w}^* \geq \gamma \\
&= \alpha\gamma G(\mathbf{w}).
\end{aligned}
$$

For the third claim, by the assumption, we have

$$\frac{1}{n} \cdot \frac{1}{1 + \exp\left(y_i f(\mathbf{w}; \mathbf{x}_i)\right)} \leq G(\mathbf{w}) \leq \frac{1}{2n},$$

which implies that

$$y_i f(\mathbf{w}; \mathbf{x}_i) \geq 0, \quad \forall i \in [n].$$

Therefore,

$$G(\mathbf{w}) = \frac{1}{n} \sum_{i=1}^{n} \frac{1}{1 + \exp\left(y_i f(\mathbf{w}; \mathbf{x}_i)\right)} \geq \frac{1}{n} \sum_{i=1}^{n} \frac{1}{2\exp\left(y_i f(\mathbf{w}; \mathbf{x}_i)\right)} = \frac{1}{2} F(\mathbf{w}).$$

We complete the proof. $\qquad\qquad\qquad\qquad\qquad\qquad\qquad\qquad\qquad\qquad\qquad\qquad\qquad\qquad\square$

The key ingredient of the phase transition analysis is the following lemma. The main idea is to consider the gradient potential $G(\mathbf{w})$ instead of the loss function $L(\mathbf{w})$ in EoS phase. And this will decrease the order of the bound of phase transition time from $\tilde{O}(\eta^2)$ to $\tilde{O}(\eta)$.

**Lemma C.2** (A bound of $\|\mathbf{w}_t\|$). *For every $\eta$, we have*

$$\|\mathbf{w}_t\| \leq \sqrt{m} \cdot \frac{2 + 8\log(\gamma^2\eta t)/\alpha + 8\kappa/\alpha + 4\eta}{\gamma} + 2\|\mathbf{w}_0\|.$$

*Proof of Lemma C.2.* By Theorem B.3, we have

$$\frac{\|\mathbf{w}_t - \mathbf{u}\|^2}{2m\eta t} \leq \frac{\|\mathbf{w}_t - \mathbf{u}\|^2}{2m\eta t} + \frac{1}{t}\sum_{k=0}^{t-1} L(\mathbf{w}^{(k)}) \leq \frac{1 + 8\log^2(\gamma^2\eta t)/\alpha^2 + 8\kappa^2/\alpha^2 + \eta^2}{\gamma^2\eta t} + \frac{\|\mathbf{w}_0\|^2}{m\eta t}.$$

Besides, we know that

$$\|\mathbf{u}\|^2 \leq 2\|\mathbf{u}_1\|^2 + 2\|\mathbf{u}_2\|^2 = \frac{4m\log(\gamma^2\eta t)^2 + 4m\kappa^2}{\alpha^2\gamma^2} + \frac{m\eta^2}{2\gamma^2}.$$

Combining them, we have

$$\|\mathbf{w}_t\|^2 \leq 2\|\mathbf{w}_t - \mathbf{u}\|^2 + 2\|\mathbf{u}\|^2 \leq m \cdot \frac{2 + 24\log^2(\gamma^2\eta t)/\alpha^2 + 24\kappa^2/\alpha^2 + 3\eta^2}{\gamma^2} + 2\|\mathbf{w}_0\|^2.$$

Hence, we can get a bound for $\|\mathbf{w}_t\|$.

$$\|\mathbf{w}_t\| \leq \sqrt{m} \cdot \frac{2 + 8\log(\gamma^2\eta t)/\alpha + 8\kappa/\alpha + 4\eta}{\gamma} + 2\|\mathbf{w}_0\|.$$

$\square$

**Lemma C.3** (Gradient potential bound in the EoS phase). *For every $\eta$, we have*

$$\frac{1}{t}\sum_{k=0}^{t-1} G(\mathbf{w}^{(k)}) \leq \frac{\langle \mathbf{w}_t, \bar{\mathbf{w}}_* \rangle - \langle \mathbf{w}_0, \bar{\mathbf{w}}_* \rangle}{m\alpha\gamma\eta t} \leq \frac{\sqrt{m}\|\mathbf{w}_t\| - \langle \mathbf{w}_0, \bar{\mathbf{w}}_* \rangle}{m\alpha\gamma\eta t}, \quad t \geq 1.$$

*Additionally, we have*

$$\frac{1}{t}\sum_{k=0}^{t-1} G(\mathbf{w}^{(k)}) \leq \frac{2 + 8\log\left(\gamma^2\eta t\right)/\alpha + 8\kappa/\alpha + 4\eta}{\alpha\gamma^2\eta t} + \frac{3\|\mathbf{w}_0\|}{\alpha\gamma\eta t}, \quad t \geq 1.$$

*This*

*Proof.* This is from the perceptron argument [Novikoff, 1962]. Specifically,

$$\langle \mathbf{w}_{t+1}, \bar{\mathbf{w}}_* \rangle = \langle \mathbf{w}_t, \bar{\mathbf{w}}_* \rangle - m\eta\langle \nabla L(\mathbf{w}_t), \bar{\mathbf{w}}_* \rangle$$

$$= \langle \mathbf{w}_t, \bar{\mathbf{w}}_* \rangle - \eta\sum_{i=1}^{n}\sum_{k=1}^{m} a_k^2 \ell'(y_i f(\mathbf{w}_t; \mathbf{x}_i)) y_i \phi(\mathbf{x}_i^\top \mathbf{w}_t^{(k)})\langle \mathbf{x}_i, \mathbf{w}_* \rangle$$

$$\geq \langle \mathbf{w}_t, \bar{\mathbf{w}}_* \rangle - \eta\sum_{i=1}^{n}\sum_{k=1}^{m} a_k^2 \ell'(y_i f(\mathbf{w}_t; \mathbf{x}_i))\alpha\gamma$$

$$\geq \langle \mathbf{w}_t, \bar{\mathbf{w}}_* \rangle + m\alpha\gamma\eta G(\mathbf{w}_t).$$

Telescoping the sum, we have

$$\frac{1}{t}\sum_{k=0}^{t-1} G(\mathbf{w}^{(k)}) \leq \frac{\langle \mathbf{w}_t, \bar{\mathbf{w}}_* \rangle - \langle \mathbf{w}_0, \bar{\mathbf{w}}_* \rangle}{m\alpha\gamma\eta t}$$

$$\leq \frac{\sqrt{m}\|\mathbf{w}_t\| - \langle \mathbf{w}_0, \bar{\mathbf{w}}_* \rangle}{m\alpha\gamma\eta t}$$

$$\leq \frac{2 + 8\log\left(\gamma^2\eta t\right)/\alpha + 8\kappa/\alpha + 4\eta}{\alpha\gamma^2\eta t} + \frac{3\|\mathbf{w}_0\|}{\sqrt{m}\alpha\gamma\eta t}. \qquad \text{by Lemma C.2}$$

We have completed the proof. $\square$

Besides, we can make use of the equivalence between $G$ and $L$ to get a bound for the loss function which is independent of the initial margin at $s$.

**Lemma C.4** (A risk bound in the stable phase for Two-layer NN). *Suppose that there exists a time $s$ such that*

$$L(\mathbf{w}_s) \leq \min\left\{\frac{1}{\eta(4+2\tilde{\beta})}, \frac{1}{2e^{\kappa+2}n}\right\}.$$

*Then for every $t \geq s+1$, we have*

$$L(\mathbf{w}_t) \leq \frac{2}{(t-s)\alpha^2\gamma^2}.$$

*Proof.* By Lemma A.3 and $f(x)$ is $\frac{1}{\sqrt{m}}$ Lipschitz and $\frac{\tilde{\beta}}{m}$ smooth, we have

$$L_{k+1} \leq L_k - m\eta(1 - (2+\tilde{\beta})\eta L(\mathbf{w}_k))\|\nabla L_t\|^2.$$

By Lemma C.1 and $L_t \leq \frac{1}{\eta(4+2\tilde{\beta})}$, we have

$$L_{k+1} \leq L_k - \frac{\alpha^2\gamma^2}{2}L_k^2.$$

Multiplying $\frac{1}{L_k^2}$ in both sides, we have

$$\frac{\alpha^2\gamma^2}{2} \leq \frac{L_t - L_{k+1}}{L_k^2} \leq \frac{1}{L_{k+1}} - \frac{1}{L_k}.$$

Taking summation for $k = s, \ldots, t-1$, we have

$$\frac{1}{L_t} > \frac{1}{L_t} - \frac{1}{L_s} \geq \frac{(t-s)\alpha^2\gamma^2}{2} \implies L_t \leq \frac{2}{(t-s)\alpha^2\gamma^2}.$$

$\square$

At last, we will use the bound for the gradient potential to get an upper bound for the phase transition time.

## C.1    Proof of Theorem 4.1

*Proof of Theorem 4.1.* Applying Lemma C.3, we have

$$\frac{1}{\tau}\sum_{k=0}^{\tau-1} G(\mathbf{w}^{(k)}) \leq \frac{2 + 8\log\left(\gamma^2\eta\tau\right)/\alpha + 8\kappa/\alpha + 4\eta}{\alpha\gamma^2\eta\tau} + \frac{3\|\mathbf{w}_0\|}{\sqrt{m}\alpha\gamma\eta\tau}$$

$$\leq \frac{2 + 8\kappa/\alpha + 8\log(\gamma^2\tau)/\alpha + (4+8/\alpha)\eta}{\alpha\gamma^2\eta\tau} + \frac{3\|\mathbf{w}_0\|}{\sqrt{m}\alpha\gamma\eta\tau} \quad \text{since } \log(\eta) \leq \eta,$$

Let $c_1 = 4e^{\kappa+2}, c_2 = (8+4\tilde{\beta})$. Note that we have

$$\frac{2+8\kappa/\alpha}{\alpha\gamma^2\eta\tau} \leq \frac{1}{4(c_1n+c_2\eta)} \text{ if } \gamma^2\tau \geq 4(2+8\kappa)\frac{c_2\eta+c_1n}{\eta\alpha^2}$$

$$\frac{8\log(\gamma^2\tau)/\alpha}{\alpha\gamma^2\eta\tau} \leq \frac{1}{4(c_1n+c_2\eta)} \text{ if } \gamma^2\tau \geq 128\frac{c_2\eta+c_1n}{\eta\alpha^2}\log\frac{c_2\eta+c_1n}{\eta}, \text{since Lemma G.5}$$

$$\frac{(4+8/\alpha)\eta}{\alpha\gamma^2\eta\tau} \leq \frac{1}{4(c_1n+c_2\eta)} \text{ if } \gamma^2\tau \geq \frac{48}{\alpha^2}(c_2\eta+c_1n),$$

$$\frac{3\|\mathbf{w}_0\|}{m\alpha\gamma\eta\tau} \leq \frac{1}{4(c_1n+c_2\eta)} \text{ if } \gamma\tau \geq \frac{12}{\alpha}\frac{(c_2\eta+c_1n)}{\eta} \cdot \frac{\|\mathbf{w}_0\|}{\sqrt{m}}$$

and that the two conditions are satisfied because

$$\gamma^2\tau := \frac{128(1+4\kappa)}{\alpha^2}\max\left\{c_2\eta, c_1n, e, \frac{c_2\eta+c_1n}{\eta}\log\frac{c_2\eta+c_1n}{\eta}, \frac{(c_2\eta+c_1n)\|\mathbf{w}_0\|}{\eta\sqrt{m}}\right\}$$

$$\geq \max\left\{4(2+8\kappa)\frac{c_2\eta+c_1n}{\eta\alpha^2}, 128\frac{c_2\eta+c_1n}{\eta\alpha^2}\log\frac{c_2\eta+c_1n}{\eta}, \frac{48}{\alpha^2}(c_2\eta+c_1n), \frac{12}{\alpha}\frac{(c_2\eta+c_1n)}{\eta}\cdot\frac{\|\mathbf{w}_0\|}{\sqrt{m}}\right\}.$$

So there exits $s \leq \tau$ such that

$$G(\mathbf{w}_s) \leq \min\left\{\frac{1}{e^{\kappa+2}4n}, \frac{1}{\eta(8+4\tilde{\beta})}\right\}$$

Then we have $L(\mathbf{w}_s) \leq F(\mathbf{w}_s) \leq 2G(\mathbf{w}_s) \leq \left\{\frac{1}{e^{\kappa+2}2n}, \frac{1}{\eta(4+2\tilde{\beta})}\right\}$. We complete the proof. $\square$

## C.2 Proof of Corollary 4.2

*Proof of Corollary 4.2.* The main idea is to show that $\tau \leq \frac{T}{2}$. Note that by Theorem 4.1, we have

$$\tau = \frac{128(1+4\kappa)}{\alpha^2}\max\left\{c_2\eta, c_1n, e, \frac{c_2\eta+c_1n}{\eta}\log\frac{c_2\eta+c_1n}{\eta}, \frac{(c_2\eta+c_1n)}{\eta}\cdot\frac{\|\mathbf{w}_0\|}{\sqrt{m}}\right\},$$

in which expression $c_1 = 4e^{\kappa+2}$ and $c_2 = (8+4\tilde{\beta})$. We can verify that,

$$\frac{128(1+4\kappa)}{\alpha^2}c_2\eta = \frac{128(1+4\kappa)}{\alpha^2}c_2\cdot\frac{\alpha^2\gamma^2}{256(1+4\kappa)c_2}T = \frac{T}{2},$$

$$\frac{128(1+4\kappa)c_1n}{\alpha^2} \leq \frac{T}{2}.$$

Furthermore, we have $n \leq \frac{\alpha^2\gamma^2 T}{256(1+4\kappa)c_1}$. Hence,

$$\frac{c_2\eta+c_1n}{\eta} = \frac{\frac{\alpha^2\gamma^2 T}{256(1+4\kappa)}+c_1n}{\frac{\alpha^2\gamma^2 T}{256(1+4\kappa)c_2}} \leq \frac{2\cdot\frac{\alpha^2\gamma^2 T}{256(1+4\kappa)}}{\frac{\alpha^2\gamma^2 T}{256(1+4\kappa)c_2}} \leq 2c_2.$$

We get that:

$$\frac{128(1+4\kappa)}{\alpha^2}\cdot\frac{c_2\eta+c_1n}{\eta}\log\frac{c_2\eta+c_1n}{\eta} \leq 2\frac{128(1+4\kappa)}{\alpha^2}c_2\ln(2c_2) \leq \frac{128(1+4\kappa)}{\alpha^2}4c_2^2 \leq \frac{T}{2},$$

$$\frac{128(1+4\kappa)}{\alpha^2}\cdot\frac{(c_2\eta+c_1n)}{\eta}\cdot\frac{\|\mathbf{w}_0\|}{\sqrt{m}} \leq \frac{128(1+4\kappa)}{\alpha^2}\cdot 2c_2\frac{\|\mathbf{w}_0\|}{\sqrt{m}} \leq \frac{T}{2}.$$

Hence, we have $\tau \leq \frac{T}{2}$. Applying Theorem 4.1, we have

$$L(\mathbf{w}_T) \leq \frac{2}{\alpha^2\gamma^2\eta(T-\tau)} \leq \frac{4}{\alpha^2\gamma^2\eta T} \leq \frac{2048(1+4\kappa)c_2}{\alpha^4\gamma^4 T^2} = \mathcal{O}(1/T^2).$$

We have completed the proof. $\square$

## C.3 Proof of Theorem 4.3

*Proof of Theorem 4.3.* The main idea is to construct an upper bound of $\eta$ and apply the analysis in Theorem 2.2. Note that give $\mathbf{w}_0 = 0$, we have

$$f(\mathbf{w}_0; \mathbf{x}_i) = \frac{1}{m}\sum_{k=1}^{m}a_k\phi(\mathbf{x}_i^\top\mathbf{w}_0^{(k)}) = s_a\phi(0),$$

where $s_a = \sum_{k=1}^{m}a_k/m$. Therefore,

$$[\nabla L(\mathbf{w}_0)]^{(k)} = \frac{1}{2}\ell'(s_a\phi(0))\cdot\frac{a_k}{m}\phi'(0)\mathbf{x}_1 + \frac{1}{2}\ell'(s_a\phi(0))\cdot\frac{a_k}{m}\phi'(0)\mathbf{x}_2$$

$$= \frac{a_k}{m}\ell'(s_a\phi(0))\phi'(0)\frac{\mathbf{x}_1+\mathbf{x}_2}{2}$$

$$= \frac{a_k}{m}\ell'(s_a\phi(0))\phi'(0)(\gamma, \frac{\sqrt{1-\gamma^2}}{4}).$$

Let $\bar{\mathbf{x}} := \frac{1}{m}\ell'(s_a\phi(0))\phi'(0)(\gamma, \frac{\sqrt{1-\gamma^2}}{4})$, we have

$$\mathbf{w}_1^{(k)} = 0 - \eta\nabla[L(\mathbf{w}_0)]^{(k)} = -\eta a_k\bar{\mathbf{x}}.$$

Therefore,

$$f(\mathbf{w}_1;\mathbf{x}_i) = \frac{1}{m}\sum_{k=1}^{m} a_k\phi(-\mathbf{x}_i^\top(\eta a_k\bar{\mathbf{x}})).$$

We can notice that $-\mathbf{x}_1^\top\bar{\mathbf{x}} < 0$ and $-\mathbf{x}_2^\top\bar{\mathbf{x}} > 0$, when $\gamma \leq 0.1$. Furthermore, we have

$$f(\mathbf{w}_1;\mathbf{x}_1) = \frac{1}{m}\sum_{a_k=1}\phi(-\mathbf{x}_1^\top(\eta\bar{\mathbf{x}})) + \frac{1}{m}\sum_{a_k=-1} -\phi(\mathbf{x}_1^\top(\eta\bar{\mathbf{x}}))$$

$$= \frac{1}{m}\sum_{a_k=1}[\phi(0) - \mathbf{x}_1^\top(\eta\bar{\mathbf{x}})\phi'(\epsilon_1)] + \frac{1}{m}\sum_{a_k=-1}[-\phi(0) - \mathbf{x}_1^\top(\eta\bar{\mathbf{x}})\phi'(\epsilon_2)]$$

$$= s_a\phi(0) - \eta\mathbf{x}_1^\top\bar{\mathbf{x}}\frac{1}{m}[\sum_{a_k=1}\phi'(\epsilon_1) + \sum_{a_k=-1}\phi'(\epsilon_2)]$$

$$\leq s_a\phi(0) - \eta\mathbf{x}_1^\top\bar{\mathbf{x}}\alpha \qquad\qquad\qquad \phi'(\epsilon_i) \geq \alpha.$$

Note that

$$\frac{1}{2}\ell(s_a\phi(0) - \eta\mathbf{x}_1^\top\bar{\mathbf{x}}\alpha) \leq \frac{1}{2}\ell(f(\mathbf{w}_1;\mathbf{x}_1)) \leq L(\mathbf{w}_1) \leq L(\mathbf{w}_0) = \ell(s_a\phi(0)).$$

We apply Lemma G.7 to get

$$\eta \leq \frac{|s_a\phi(0)| + \ln 3}{\mathbf{x}_1^\top\bar{\mathbf{x}}\alpha}.$$

We use $c_3 := \frac{|s_a\phi(0)| + \ln 3}{\mathbf{x}_1^\top\bar{\mathbf{x}}\alpha}$. Now we know $\eta \leq c_3$. Furthermore, notice that

$$\|\nabla L(\mathbf{w}_t)\| \leq L_t \leq L_0.$$

We get that $\|\mathbf{w}_{t+1} - \mathbf{w}_t\| \leq \eta L_0 \leq c_3 L_0$. Hence,

$$|f(\mathbf{w}_{t+1};\mathbf{x}_i) - f(\mathbf{w}_t;\mathbf{x}_i)| \leq c_3 L_0.$$

Assume that $l_b = \min\left\{\frac{1}{e^{\kappa+2}4n}, \frac{1}{\eta(8+4\tilde{\beta})}\right\}$ and

$$L_{s-1} \geq l_b, \quad L_s \leq l_b.$$

We know that

$$l_b \geq \min\left\{\frac{1}{e^{\kappa+2}4n}, \frac{1}{c_3(8+4\tilde{\beta})}\right\} =: l_c.$$

We want to show that there is an lower bound for $L_s$. Now that

$$L_s = \frac{1}{2}\Big[\ell(f(\mathbf{w}_s;\mathbf{x}_1)) + \ell(f(\mathbf{w}_s;\mathbf{x}_2))\Big].$$

Applying Lemma G.6, we can get that

$$L_s \geq \exp(-c_3 L_0)L_{s-1} \geq \exp(-c_3 L_0)l_b.$$

Recall that by Lemma A.14, we have

$$L_t \geq \frac{1}{\frac{1}{L_s} + 3\tilde{\eta}\rho^2(t-s)}, \quad t \geq s.$$

Combine this with $\rho = \frac{1}{\sqrt{m}}, \tilde{\eta} = \eta m$ and $L_s \geq \exp(-c_3 L_0)l_b$ and we get

$$L_t \geq \frac{1}{\frac{\exp(c_3 L_0)}{l_b} + 3\eta(t-s)}, \quad t \geq s.$$

Note that when $t \leq s$, $L_t \geq l_b$. We can get a lower bound for $L_t$ by

$$L_t \geq \frac{1}{\frac{\exp(c_3 L_0)}{l_b}t + 3\eta t} \geq \frac{1}{\frac{\exp(c_3 L_0)}{l_b}t + 3c_3 t} \geq \frac{1}{\frac{\exp(c_3 L_0)}{l_c}t + 3c_3 t} = \frac{c_4}{t},$$

where $c_4 = \frac{1}{\frac{\exp(c_3 L_0)}{l_c} + 3c_3}$ depends only on $\{a_j\}_{j=1}^m, \phi(0), \kappa, \tilde{\beta}$ and $n$. $\qquad\square$

# D  Scaling and Homogenous Error

In this section, we consider different scaling of two-layer networks. We add a scaling factor $b$ into the model, i.e.,

$$f(\mathbf{w}; \mathbf{x}) = \frac{b}{m} \sum_{j=1}^{m} a_j \phi(\mathbf{x}^\top \mathbf{w}^{(j)}).$$

We will show that given a limited computation budget $T$ (total iterations), larger $b$ and a corresponding best $\tilde{\eta} = \eta \cdot m$ will achieve the same best rate as $b = 1$, i.e., $O(1/T^2)$. While for smaller $b$, the rate is $O(b^{-3}/T^2)$. Before we present the analysis, here are the bounds with $b$ and $\tilde{\eta} = m \cdot \eta$ following the process of Lemma C.3:

$$\frac{1}{t} \sum_{k=0}^{t-1} L(\mathbf{w}_k) \leq \frac{1 + 8\log^2(\gamma^2 \eta t)/(\alpha^2 b^2) + 8\kappa^2/\alpha^2 + \eta^2 b^2}{\gamma^2 \eta t} + \frac{\|\mathbf{w}_0\|^2}{m\eta t},$$

$$\frac{1}{t} \sum_{k=0}^{t-1} G(\mathbf{w}_k) \leq \frac{2 + 8\log(\gamma^2 \eta t)/(\alpha b) + 8\kappa/\alpha + 2\eta b}{\alpha\gamma^2 b\eta t} + \frac{3\|\mathbf{w}_0\|}{\sqrt{m}\eta bt}.$$

**Case when $b \geq 1$.**  Given the previous bounds, we have the following results following the idea in Appendix C:

- Gradient potential bound: $G(\mathbf{w}_t) \leq \frac{C}{t}$ for all $t \geq 0$,

- Phase transition threshold: $G(\mathbf{w}_s) \leq \min\left\{1/4e^{\kappa+2}n, 1/\eta(8\rho^2 b^2 + 4\tilde{\beta}b)\right\}$,

- Stable phase bound: $L(\mathbf{w}_t) \leq \frac{2}{Cb^2\eta(t-s)}$,

where $C$ depends on $\alpha, \gamma$. Combine the first two arguments and assume $\eta(8\rho^2 b^2 + 4\tilde{\beta}b) \geq 4e^{\kappa+2}n$. We get $s \leq C\eta(8\rho^2 b^2 + 4\tilde{\beta}b)$. Plug this into the third bound. We have

$$L(\mathbf{w}_T) \leq \frac{2}{Cb^2\eta(T - C\eta(8\rho^2 b^2 + 4\tilde{\beta}b))}.$$

It's obvious that the best $\eta = \frac{T}{16\rho^2 b^2 C + 8\tilde{\beta}bC}$. Hence,

$$L(\mathbf{w}_T) \leq \frac{8(8\rho^2 b^2 C + 4\tilde{\beta}bC)}{Cb^2 T^2} = \mathcal{O}\left(\frac{1}{T^2}\right).$$

Then, the rate is still $O(1/T^2)$.

**Case when $b < 1$.**  Similarly, we can get the following bounds:

- Gradient potential bound: $G(\mathbf{w}_t) \leq \frac{Cb^{-2}}{t}$ for all $t \geq 0$,

- Phase transition threshold: $G(\mathbf{w}_s) \leq \min\left\{1/4e^{\kappa+2}n, 1/\eta(8\rho^2 b^2 + 4\tilde{\beta}b)\right\}$,

- Stable phase bound: $L(\mathbf{w}_t) \leq \frac{2}{Cb^2\eta(t-s)}$,

where $C$ depends on $\alpha, \gamma$. Without loss of generality, we can assume $\eta(8\rho^2 b^2 + 4\tilde{\beta}b) \geq 4e^{\kappa+2}n$, since $\eta$ can be small enough. Then, we have

$$s \leq C\eta(8\rho^2 + 4\tilde{\beta}b^{-1}).$$

Then, we can get

$$L(\mathbf{w}_T) \leq \frac{2}{Cb^2\eta(T - C\eta(8\rho^2 + 4\tilde{\beta}b^{-1}))}.$$

It's obvious that the best $\eta = \frac{T}{16\rho^2 C + 8\tilde{\beta}b^{-1}C}$. Hence,

$$L(\mathbf{w}_T) \leq \frac{8(8\rho^2 Cb^{-2} + 4\tilde{\beta}b^{-3}C)}{CT^2} = \mathcal{O}\left(\frac{b^{-3}}{T^2}\right).$$

Combining the analysis for two cases, we observe that when $b \geq 1$, the fast loss rate is $\mathcal{O}(1/T^2)$ given finite budget $T$. While $b < 1$, the rate is $\mathcal{O}(b^{-3}/T^2)$. In our main results, we set $b = 1$ for the mean-field scaling. Under the mean-field regime, all bounds are independent of the number of neurons since we consider the dynamics of the distributions of neurons. Alternatively, if we set $b = \sqrt{m}$, then the model becomes:

$$f(\mathbf{w}; \mathbf{x}) = \frac{1}{\sqrt{m}} \sum_{j=1}^{m} a_j \phi(\mathbf{x}^\top \mathbf{w}^{(j)}).$$

The model falls into the NTK regime. The loss threshold will be related to $m$, but the loss rate is the same as that of the mean-field scaling.

# E    Extra Experiments

Here we provide additional experiments to support our theoretical results. In Figure 3, we show the test accuracy of two-layer networks for CIFAR-10 under the same setting of Figure 2. We can observe that large stepsizes lead to stronger implicit biases with "nicer" features.

In Figure 4, we show the training loss and margins of a two-layer network with leaky softplus activations on a synthetic linear separable dataset. We can observe that both neurons have negative margins during the training, while the network's margin increases and becomes positive. This indicates that even the two-layer networks can have complicated dynamics. It remains an open question to understand each neuron's dynamics in deep networks.

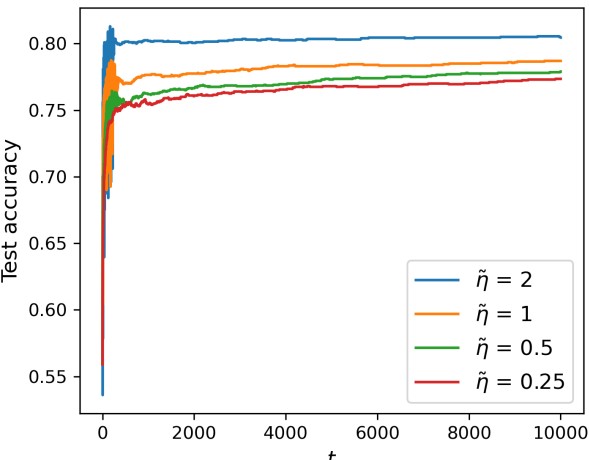

Figure 3: Test accuracy of two-layer networks for CIFAR-10 under the same setting of Figure 2(d)-(f).The results support our intuition that large stepsizes lead to stronger implicit biases with "nicer" features.

# F    Additional Proofs

## F.1    Proof of Example 2.1

*Proof of Example 2.1.* Recall that the two-layer neural network is defined as:

$$f(\mathbf{w}; \mathbf{x}) = \frac{1}{m} \sum_{j=1}^{m} a_j \phi(\mathbf{x}^T \mathbf{w}^{(j)}).$$

We can verify that if $\phi(x)$ is $\beta$-smooth and $\rho$-Lipschitz with respect to $x$, then $f(\mathbf{w}; \mathbf{x})$ is $\beta/m$-smooth and $\rho/\sqrt{m}$-Lipschitz with respect to $\mathbf{w}$. This is because:

$$\nabla L(\mathbf{w}) = \hat{\mathbb{E}}\ell'(yf(\mathbf{w}; \mathbf{x}))y\nabla f(\mathbf{w}; \mathbf{x}),$$

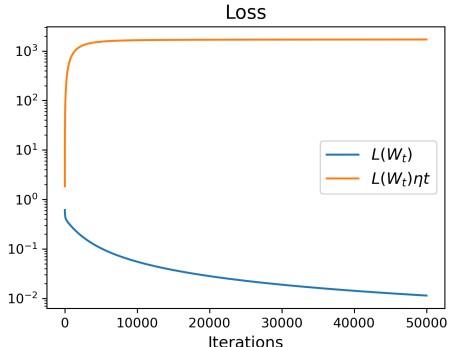

(a) Training loss, synthetic dataset.

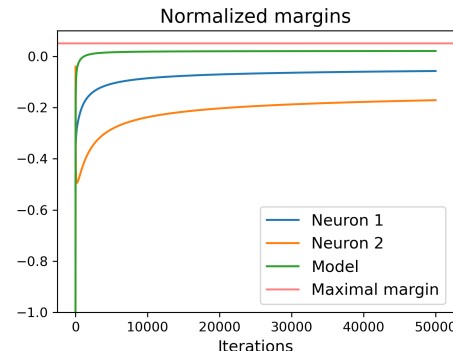

(b) Normalized Margins, synthetic dataset.

Figure 4: Training loss and margins of a two-layer network with leaky softplus activations on a synthetic linear separable dataset. There are five samples in the dataset, which are $((0.05, 1, 2), 1), ((0.05, -2, 1), 1), ((-1, 0, 2), -1), ((0.05, -2, -2), 1), ((0.05, 1, -2), 1)$. The max margin direction is $(1, 0, 0)$ with a normalized margin of $0.05$. The network only has two neurons with fixed weights $1/2$ and $-1/2$. The leaky softplus activation is $\tilde{\phi}(x) = (x + \phi(x))/2$, where $\phi$ is the softplus activation. The stepsize is 3. We can observe that both neurons have negative margins during the training, while the network's margin increases and becomes positive.

$$\nabla^2 L(\mathbf{w}) = \hat{\mathbb{E}} \ell''(y f(\mathbf{w}; \mathbf{x})) \nabla f(\mathbf{w}; \mathbf{x})^{\otimes 2} + \ell'(y f(\mathbf{w}; \mathbf{x})) y \nabla^2 f(\mathbf{w}; \mathbf{x}),$$

and that

$$\nabla f(\mathbf{w}; \mathbf{x}) = \begin{pmatrix} \vdots \\ \frac{1}{m} a_j \phi'(\mathbf{x}^\top \mathbf{w}^{(j)}) \mathbf{x} \\ \vdots \end{pmatrix}, \quad \nabla^2 f(\mathbf{w}; \mathbf{x}) = \begin{pmatrix} \ddots & & 0 & & 0 \\ 0 & \frac{1}{m} a_j \phi''(\mathbf{x}^\top \mathbf{w}^{(j)}) \mathbf{x}\mathbf{x}^\top & 0 \\ 0 & & 0 & & \ddots \end{pmatrix}.$$

Now, we will focus on the parameters of each activation function.

- **GELU**. $\phi(x) = x \cdot \mathrm{erf}(1 + (x/\sqrt{2}))/2 = x \cdot F(x)$.

$$\phi'(x) = F(x) + x \cdot f(x),$$
$$\phi''(x) = 2f(x) + x \cdot f'(x),$$

where $F(x), f(x)$ are the CDF and PDF of standard normal distribution. Note that $x f(x) = \frac{x}{\sqrt{2\pi}} e^{-x^2/2}$ and $(x f(x))' = \frac{1}{\sqrt{2\pi}}(1 - x^2) e^{-x^2/2}$. We can find the maximum of $x f(x)$ is $\frac{1}{\sqrt{2\pi}} e^{-1/2}$. Besides, we know that $F(x), f(x) \leq 1$ and $x \cdot f'(x) \leq 0$. Combining them, we have $\rho = 1 + e^{-1/2}/\sqrt{2\pi}$ and $\beta = 2$. For $\kappa$, $\phi - \phi'(x)x = -x \cdot f(x)$. So the bound of $\kappa$ is $e^{-1/2}/\sqrt{2\pi}$.

- **Softplus**. $\phi(x) = \log(1 + e^x)$. Therefore,

$$\phi'(x) = \frac{e^x}{1 + e^x} \leq 1,$$
$$\phi''(x) = \frac{e^x}{(1 + e^x)^2} \leq 1.$$

Besides,

$$(\phi(x) - \phi'(x)x)' = \left( \log(1 + e^x) - \frac{e^x x}{1 + e^x} \right)' = -\frac{e^x x}{(1 + e^x)^2}.$$

So the maximum is $\phi(0) - \phi'(0)0 = \log 2$. Besides, when $x > 1$, $\phi(x) \geq x \geq \frac{e^x x}{1 + e^x}$. When $x \to -\infty$, $\phi(x) - \phi'(x)x \to 0$. Therefore, $\kappa = \log 2$.

- **Sigmoid**. $\phi(x) = 1/(1 + e^{-x})$. Hence,

$$\phi'(x) = \frac{e^{-x}}{(1 + e^{-x})^2} \le 1,$$

$$\phi''(x) = \frac{e^{-2x} - e^{-x}}{(1 + e^{-x})^3} \le 1.$$

As for $\kappa$, we know that

$$|\phi(x) - \phi'(x)x| = \left| \frac{1 + e^{-x} - xe^{-x}}{(1 + e^{-x})^2} \right| \le \frac{1 + e^{-x} + |x|e^{-x}}{(1 + e^{-x})^2}.$$

Note that $|x|e^{-x} \le e^{-2x} + 1$. We have

$$|\phi(x) \le \frac{1 + e^{-x} + e^{-2x} + 1}{(1 + e^{-x})^2} \le 2.$$

- **Tanh**. $\phi(x) = \frac{e^x - e^{-x}}{e^x + e^{-x}} \le 1$. Note that

$$\phi'(x) = 1 - \phi(x)^2 \le 1$$

$$\phi''(x) = 2\phi(x)^3 - \phi(x) \le 2.$$

Besides, we know that

$$|x\phi'(x)| = \frac{4|x|}{(e^x + e^{-x})^2} \le 4.$$

Hence

$$|\phi(x) - \phi'(x)| \le |\phi(x)| + |x\phi'(x)| \le 5.$$

- **SiLU**. Note that

$$\phi'(x) = \frac{1 + e^{-x} + xe^{-x}}{(1 + e^{-x})^2},$$

$$\phi''(x) = \frac{(2 - x)e^{-x}}{(1 + e^{-x})^2} + \frac{xe^{-2x}}{(1 + e^{-x})^3}.$$

Because $|x|e^{-x} \le e^{-2x} + 1$ and $|x|e^{-2x} \le e^{-3x} + 1$. We get $|\phi'(x)| \le 2$ and $\phi''(x)| \le 4$. At last,

$$|\phi(x) - x\phi'(x)| = \frac{|x|e^{-x}}{(1 + e^{-x})^2} \le 1.$$

- **Huberized ReLU**. It's obvious that $\phi'(x) \le 1$ and $\beta = 1/h$. Note that $\phi$ is not second-order differentiable. At last,

$$|\phi(x) - x\phi'(x)| = \begin{cases} 0 & x < 0, \\ x^2/2h & 0 \le x \le h, \\ h/2 & x > h. \end{cases}$$

Hence, it's upper bounded by $h/2$.

$\square$

### F.2 Proof of Example 3.1

*Proof of Example 3.1.* Because for activation functions in Example 2.1, $\beta \le 4$ and $\rho \le 2$. Hence, for $\tilde{\phi}(x) = cx + (1 - c)\phi(x)/4$, $\tilde{\beta} = 1$ and $\rho = 1$. Besides, since $0.5 < c < 1$, we must have $(\tilde{\phi}(x))' \ge 0.25$. $\square$

# G   Additional Lemmas

**Lemma G.1.** *If $\frac{1}{2} \geq L_1 \geq \frac{1}{c}$ and $L_2 \geq L_1 - L_1^2$, we have*

$$L_2 \geq \frac{1}{c+2}.$$

*Proof.* For function $g(x) = x - x^2$, $g'(x) = 1 - 2x$. If $x \leq \frac{1}{2}$, then $g(x)$ is increasing. Then

$$g(L_1) \geq g(\frac{1}{c}) = \frac{c-1}{c^2} = \frac{c^2 + c - 2}{c^2(c+2)} \geq \frac{1}{c+2}.$$

$\square$

**Lemma G.2.** *Given a continuous function $f(x)$ s.t. $|f(x) - \langle \nabla f(x), x \rangle| \leq \kappa$, then for a fixed constant $r > 0$ there exists $C_{r,\kappa}$ and $C_r$ s.t. for any $\|x\| \geq r$,*

$$|f(x)| \leq C_{r,\kappa} \|x\|,$$

*and for any $x$,*

$$|f(x)| \leq C_{r,\kappa} \|x\| + C_r.$$

*Proof.* Since $f$ is continuous, let

$$C_r = \max_{\|x\|=r} |f(x)|/r.$$

Now for any $\|x\| > r$, let $y = \frac{rx}{\|x\|}$ and consider $g(s) = \frac{f(sy)}{s}$. Then we have

$$g'(s) = \frac{\langle \nabla f(sy), sy \rangle - f(sy)}{s^2}.$$

Therefore, $-\frac{\kappa}{s^2} \leq g'(s) \leq \frac{\kappa}{s^2}$. Let $s = \|x\|/r$,

$$\frac{f(x)r}{\|x\|} = g(s) = g(1) + \int_1^s g'(t)dt$$

$$\leq g(1) + \int_1^s \frac{\kappa}{t^2}dt \leq g(1) + \kappa$$

$$\leq rC_r + \kappa.$$

Therefore, $f(x) \leq (C_r + \frac{\kappa}{r}) \cdot \|x\|$. Similarly, we can show that $-f(x) \leq (C_r + \frac{\kappa}{r}) \cdot \|x\|$. Therefore, for any $\|x\| \geq r$,

$$|f(x)| \leq (C_r + \frac{\kappa}{r}) \cdot \|x\|.$$

Let $D = \max_{\|x\| \leq r} |f(x)|$, we have for any $x$,

$$|f(x)| \leq (C_r + \frac{\kappa}{r}) \cdot \|x\| + D.$$

We have completed the proof.

$\square$

**Lemma G.3.** *Fixing $c > 1$, then for every $0 < x \leq \frac{1}{c}$, we have*

$$x \leq \frac{-1}{\log(cx)}.$$

*Proof of Lemma G.3.* This is equivalent to show that

$$x \log(cx) \geq -1.$$

Let $s(x) = x \log(cx)$, then $s'(x) = 1 + \log(cx)$. Hence $s(x)$ is decreasing when $0 < x < \frac{1}{ce}$ and is increasing when $x \geq \frac{1}{ce}$. The minimum of $s(x)$ is achieved at $x = \frac{1}{ce}$, which is

$$s(1/(ce)) = -\frac{1}{ce} \geq -1.$$

This completes the proof.

$\square$

**Lemma G.4.** *Given $0 < b \leq \frac{1}{2}$ and $a > 0$, we have*

$$\frac{1+a}{1-b} \leq (1+2a+2b).$$

*Proof.* This is equivalent to show that

$$(1+a) \leq (1+2a+2b)(1-b) = 1 + 2a + b - 2ab - 2b^2.$$

This is equivalent to

$$2b(a+b) \leq (a+b).$$

Since $a + b > 0$ and $b \leq \frac{1}{2}$, this is true. □

**Lemma G.5.** *Given $c > e$, we have for any $x > 2c \log c$,*

$$\frac{\log x}{x} \leq \frac{1}{c}.$$

*Proof.* It's equivalent to show that $x - c \log x \geq 0$. Let $g(x) = x - c \log x$. $g'(x) = 1 - c/x$. When $x > 2c \log c > 2c$, $g'(x) < 0$. Hence, the minimal is $g(2c \log c)$. Note that

$$g(2c \log c) = 2c \log c - c \log c - c \log 2 - c \log \log c = c \log c - c \log 2 - c \log \log c = c \log \frac{c}{2 \log c}.$$

Now we want to show that $c > 2 \log c$. Let $h(y) = y - 2 \log y$. $h'(y) = 1 - 2/y > 0$ when $y > e$. $h(e) = e - 2 > 0$. Hence $h(c) > h(e) > 0$ and $g(2c \log c) > 0$. This leads to $g(x) > 0$. Then, we complete the proof. □

**Lemma G.6.** *Given $\ell(x) = \log(1 + e^{-x})$ and $c > 0$, we have for any $x$,*

$$\ell(x + c) \geq \exp(-c)\ell(x).$$

*Proof.* Let $g(x) = \ell(x + c) - \exp(-c)\ell(x)$. Then, we have

$$g'(x) = \frac{-1}{1 + \exp(x+c)} + \frac{1}{\exp(c) + \exp(x+c)} < 0.$$

Therefore, $g(x)$ is monotonically decreasing. When $x \to \infty$, we have

$$\lim_{x \to \infty} g(x) = \lim_{x \to \infty} [\ell(x+c) - \exp(-c)\ell(x)] = \exp(-x-c) - \exp(-c)\exp(-x) = 0.$$

Therefore, $g(x) \geq 0$ for any $x$. Now, we complete the proof. □

**Lemma G.7.** *Assume $\ell(x) = \log(1 + e^{-x})$. If $\ell(x + c) \leq 2\ell(x)$, we have*

$$c \leq \ln 3 + |x|.$$

*Proof.* Note that

$$\ell(x + c) - 2\ell(x) = \log \frac{1 + e^{x+c}}{1 + 2e^x + e^{2x}} \leq 0.$$

Then,

$$\frac{1 + e^{x+c}}{1 + 2e^x + e^{2x}} \leq 1 \implies e^c \leq 2 + e^x \leq 2 + e^{|x|} \leq 3e^{|x|}.$$

Therefore, $c \leq \ln 3 + |x|$. □

