# OpenReview forum: "Large Stepsize Gradient Descent for Non-Homogeneous Two-Layer Networks: Margin Improvement and Fast Optimization"
_NeurIPS.cc/2024/Conference — NeurIPS 2024 poster_

### Official Review · Reviewer_cshA · 2024-06-23

**Soundness:** 3
**Presentation:** 4
**Contribution:** 3
**Rating:** 6
**Confidence:** 4

**Summary:**

This paper studies the effect of large learning rates for near-homogeneous logistic regressions. Especially, it extends previous results on linear predictors under large learning rates (an EoS phase and a stable phase for the convergence of GD; faster convergence) to nonlinear predictors with Lipschitzness and Lipschitz smoothness, for example, the two-layer neural networks with last layer fixed. The authors also prove the margin improvement for large learning rates and this near-homogeneous model. In the end, they demonstrate a faster convergence using large learning rates.

**Strengths:**

1. This work is the first to prove the margin improvement for large learning rates and this near-homogeneous model.
2. The authors analyze a nonlinear (near-homogeneous) model, which is closer to practice than previous works.

**Weaknesses:**

The role and effect of large learning rates, especially compared to small learning rates, may need more illustration.

**Questions:**

1. What is the initial condition in Theorem 3.2?
2. For the margin improvement part (Theorem 2.2), it would be clearer if the dependence on the learning rates in the modified margin function could be explained in the main body. Also, it would be good if there were direct comparisons between small and large learning rates results of margin improvement, e.g., different dependence on learning rates or other parameters. This may help better illustrate the advantages of using large learning rates.

**Limitations:**

Although the authors improve the theory from linear models to near-homogeneous models, it is still far from the ones used in practice. If possible, some discussions on the intuitions of more nontrivial nonlinear models may be helpful.

---

> ### Author Rebuttal · Authors · 2024-08-06
>
> We appreciate your positive feedback. We answer your questions as follows.
>
> ---
>
> **Q1**. The role and effect of large learning rates, especially compared to small learning rates, may need more illustration.
>
> **A1**. Corollary 4.2 and Theorem 4.3 together show a separation between large and small stepsize, where a large stepsize enables $O(1/T^2)$ loss, but small stepsizes suffer from at least a $\Omega(1/T)$ loss. This illustrates the effect of large stepsize for fast optimization.
>
> ---
>
> **Q2**. What is the initial condition in Theorem 3.2?
>
> **A2**. We do not have explicit assumptions on the initial condition of $w_0$. For the bound in Theorem 3.2 to be non-vacuous, $|| w_0 ||$ needs to be small compared to $\eta t$.
>
> ---
>
> **Q3**.  For the margin improvement part (Theorem 2.2), it would be clearer if the dependence on the learning rates in the modified margin function could be explained in the main body. Also, it would be good if there were direct comparisons between small and large learning rates results of margin improvement, e.g., different dependence on learning rates or other parameters. This may help better illustrate the advantages of using large learning rates.
>
> **A3**. The modified margin is defined in Equation (10) in the appendix with explicit dependence on the stepsize. We will add this to the main paper.
>
> The impact of the stepsize on the margin improvement is complex. Let us use $s$ to denote the starting time of the stable phase. If we assume $w_s$ is independent of the stepsize, then the loss and the parameter norm depend on the stepsize through the bounds $\Theta(1/\eta(t-s))$ and $\Theta(\log(\eta(t-s)))$, respectively. However, $w_s$ depends on the stepsize through a complex function, making it difficult to determine the quantitative effect of stepsize on margin improvement.
>
> ---
>
> **A4**. Although the authors improve the theory from linear models to near-homogeneous models, it is still far from the ones used in practice. If possible, some discussions on the intuitions of more nontrivial nonlinear models may be helpful.
>
> **Q4**. We believe our intuitions about the transition from an EoS phase to a stable phase extend to more general nonlinear models. Specifically, the initial EoS phase happens when GD oscillates within a sharp valley, and GD enters the stable phase when it navigates itself into a flat valley. Our theory of large stepsize is consistent with the celebrated flat minima intuition. We will add this discussion in the revision.
>
> ---

---

> > ### Comment · Reviewer_cshA · 2024-08-11
> >
> > I would like to thank the authors for the rebuttal. Please consider adding the discussions on the initial condition in Theorem 3.2 to the paper. I will maintain my score.

---

### Official Review · Reviewer_PHJ6 · 2024-07-03

**Soundness:** 3
**Presentation:** 3
**Contribution:** 2
**Rating:** 6
**Confidence:** 4

**Summary:**

This work analyzes dynamics of large step-size for GD under logistic loss for non-homogenous two layer networks. They characterize two phases, first in which empirical risk oscillates and then monotonically decreases in the second phase. Additionally, they show
1) normalized margin grows nearly monotonically in the second phase
2) in the oscillatory EOS phase the average empirical risk decreases under certain conditions
3) With a larger step-size, the GD undergoing phase transition is more efficient.

**Strengths:**

1) The paper is well written and builds on top of exisitng works on EOS for logistic loss https://arxiv.org/pdf/2305.11788 and https://arxiv.org/pdf/2402.15926 and extends these two works from linear network to two layer non-linear network.
2) This paper analyzes the large learning rate regime with non-homoegenous two layer networks (for logistic loss). This is a practical setting since most works study gradient flow or linear overparameterized models.

**Weaknesses:**

1) I liked the way the authors introduced the three model conditions to extend their result from Theorem-1 in https://arxiv.org/pdf/2402.15926 in terms of the three constants (Lipschitzness, smoothness and near-homogenity). However, these conditions do not hold for Relu network (assumption 1.B and 1.C breaks), which probably is most used non-linear activation out there. Infact there are a bunch of non-differentiable activation functions for which these assumptions do not hold as well as the final result.
2) By comparing the proofs of this work with that of https://arxiv.org/pdf/2402.15926 , it seems like similar proof technique can be utilized to incorporate the three assumptions to derive the same results involving the three constants.

**Questions:**

1) Will the loss landscape with non-homogenous layers be the same as the linear networks case? For example, the authors show that in logistic regression, the landscape is a valley instead of a quadratic basin. The iterates converges quickly in max-margin subspace direction and the oscillations within the orthogonal direction becomes small progressively with time. Can you give an intuition of how the loss landscape may look for non-homogenous layers?
2) Is it true that unllike MSE losses, where there is a chance of divergence for very high lrs, the logistic regression losses always converge irrespective of how large lr is used.
3) I missed the part in the paper where the authors discuss progressive sharpening. Usually for MSE losses, PS first takes place unless the sharpness hits 2/lr after which loss oscillates around walls of basin. Does logistic regression loss do not exhibit PS? Seems like EOS starts from the offset and then oscillations stop when iterates enter a valley of flatter oscillation directions.

**Limitations:**

As discussed the analysis breaks when considering non-differentiable activation functions like RELU. The analysis presented in the paper is an extension of previous work which I mentioned with some relaxation on the non-linearity (see the three conditions). It would be better if the authors can emphasize this limitation and point out the novelty directions in this paper.

---

> ### Author Rebuttal · Authors · 2024-08-06
>
> Thank you for your feedback. We address your concerns below.
>
> ---
>
> **Q1**. However, these conditions do not hold for Relu network (assumption 1.B and 1.C breaks), which probably is most used non-linear activation out there. Infact there are a bunch of non-differentiable activation functions for which these assumptions do not hold as well as the final result.
>
> **A1**. Note that Assumption 1.C holds for ReLU networks with $\kappa=0$ when replacing the gradients with subgradients.
>
> Since our focus is non-homogeneous networks, we choose to work with differentiable predictors to simplify the analysis. While we do not cover ReLU activation, we have addressed many other commonly used activation functions, such as GeLU and SiLU. These two are especially interesting due to their non-homogeneity. Therefore, they cannot be covered by the prior theory. Given that our work is the first to prove margin improvement for non-homogenous predictors, we believe our contributions are very significant.
>
> Extending our results to non-differentiable predictors using tools from [Lyu and Li, 2020] is interesting, and we will comment on this as a future direction.
>
> ---
>
> **Q2**. By comparing the proofs of this work with that of https://arxiv.org/pdf/2402.15926 , it seems like similar proof technique can be utilized to incorporate the three assumptions to derive the same results involving the three constants.
>
> **A2**. We emphasize that our results are for nonlinear networks, but [Wu et al., 2024] only focused on linear predictors (or networks in the NTK regime). While our EoS analysis is motivated by [Wu et al., 2024], identifying the correct set of assumptions to enable the analysis for nonlinear networks requires nontrivial efforts.
>
> Besides, our stable phase analysis also proves margin improvement for non-homogenous predictors, which partially solves an open problem noted by [Ji and Telgarsky, 2020]. To prove this result, we use techniques significantly different from those of [Wu et al., 2024].
>
>
> ---
>
> **Q3**. Will the loss landscape with non-homogenous layers be the same as the linear networks case? For example, the authors show that in logistic regression, the landscape is a valley instead of a quadratic basin. The iterates converges quickly in max-margin subspace direction and the oscillations within the orthogonal direction becomes small progressively with time. Can you give an intuition of how the loss landscape may look for non-homogenous layers?
>
> **A3**. No, the loss landscape for non-homogeneous networks is significantly different from that of logistic regression. Note that the formal is non-convex while the latter is convex. It is difficult to develop geometric intuitions about the non-convex landscape of non-homogeneous networks. This difficulty also demonstrates the significance of our contributions to establishing margin improvement and EoS theory for non-homogeneous networks.
>
> ---
>
> **Q4**. Is it true that unllike MSE losses, where there is a chance of divergence for very high lrs, the logistic regression losses always converge irrespective of how large lr is used.
>
> **A4**. This is a good question and we do not know the answer. In practice, the loss can diverge when training deep networks with very large stepsizes, even under logistic loss. However, we are not sure if this is a numerical issue or if there is a provable upper bound for the convergent stepsize.
>
> ---
>
> **Q5**. I missed the part in the paper where the authors discuss progressive sharpening. Usually for MSE losses, PS first takes place unless the sharpness hits 2/lr after which loss oscillates around walls of basin. Does logistic regression loss do not exhibit PS? Seems like EOS starts from the offset and then oscillations stop when iterates enter a valley of flatter oscillation directions.
>
> **A5**. Empirically, progressive sharpening also happens under logistic loss, as shown by [Cohen et al., 2020]. However, this is beyond the scope of this work, which focuses on the implicit bias and convergence of large stepsize GD.
>
> ---

---

> > ### Comment · Reviewer_PHJ6 · 2024-08-10
> > **Reviewer response-1**
> >
> > I thank the authors for their response. Since most of the comments seem addresses, I think Q-2 could have been addressed with more details rather than "analysis for nonlinear networks requires nontrivial efforts" and "use techniques significantly different from those of [Wu et al., 2024]". I think this question is important since the nature of results overlap with those of https://arxiv.org/pdf/2402.15926. I would suggest that the authors explain or mention how different are the proof techniques from https://arxiv.org/pdf/2402.15926  atleast briefly in the manuscript.

---

> > > ### Author Response · Authors · 2024-08-10
> > > **A technical comparison with [Wu et al., 2024]**
> > >
> > > Thank you for the suggestions. We make a detailed  technical comparison with [Wu et al., 2024] below. We will add these discussions in the revision.
> > >
> > >
> > > Our stable phase analysis shows the margin improvement in non-homogenous networks. In comparison, the stable phase analysis in [Wu et al., 2024] only concerns the convergence of the loss. As margin improvement is harder to show (especially in non-homogenous cases), the techniques of [Wu et al., 2024] are insufficient to achieve our goal. To achieve our goal, we analyze the evolvements of several modified versions of the margin. None of these quantities appear in [Wu et al., 2024]. So our techniques here are significantly different from [Wu et al., 2024].
> > >
> > > Regarding our EoS and acceleration analysis, we use tools from [Wu et al., 2024], as there are not many tools that can deal with large stepsizes. Besides extending their results from linear models to networks, we make two innovations in our analysis. First, our comparator $u_1$ (see Equation 17 in Appendix B) contains an extra component to accommodate the non-homogeneity of the predictor. Second, our Lemma C.4 uses a sharper convergence analysis, which removes some logarithmic factors in Theorem 4.1 and Corollary 4.2, comparing to these results in [Wu et al., 2024]. For instance, our Corollary 4.2 gets $O(1/T\^2)$ while their Corollary 2 gets $O(\\log(T)\^2 / T\^2)$. This is mentioned in Lines 224-227, but we will emphasize it more in the revision.

---

> > > > ### Comment · Reviewer_PHJ6 · 2024-08-13
> > > >
> > > > I thank the authors for their response. I updated my score.

---

### Official Review · Reviewer_RDnn · 2024-07-11

**Soundness:** 4
**Presentation:** 3
**Contribution:** 2
**Rating:** 6
**Confidence:** 4

**Summary:**

This paper studies GD for nearly-1-homogeneous neural networks with large stepsizes. It provides two main results.


The first one describes the late, *stable phase* of training and can be seen as an extension of Lyu and Li (2020) result to the large stepsize, nearly homogeneous setting. Yet it comes with weaker conclusions, showing that if the training loss becomes smaller than some threshold at some point, then afterwards:
     - the training loss converges to 0 at a $\frac{1}{t}$ rate
     - the normalized margin "increases", and hence converges

The second one studies the early stage, *Edge of Stability* phase for linearly separable data for a more restricted one hidden layer neural network architecture (with fixed output weights) and shows that during this early stage, the loss decreases at some rate **in average**. This then allows to provide a result illustrating the advantage of large stepsize (with an optimization error of order $\frac{1}{T^2}$) vs small stepsize (optim error $\frac{1}{T}$).

**Strengths:**

The study of GD with large stepsizes is of high relevance. This paper puts in perspective the phenomenon of edge of stability and illustrates on a simple example how it can be beneficial to faster rates for the training loss. The claimed results seem sound and the paper is nicely written.

**Weaknesses:**

The provided results and settings might be a bit too weak in my opinion. As a major drawback, the authors insist a lot on the importance of large step sizes for implicit bias in the paper. However, the only implicit bias result is claiming the normalized margin is increasing in the late stable phase. As commented by the authors in the paper, Lyu and Li provided a KKT point convergence result in their work, and I was hoping the same conclusion could be possible here. As a consequence, this paper mostly studies the convergence rates of the training loss, which I find less interesting from a personal point of view.
Actually, discrepancies in the normalized margins can be seen in the different experiments of the paper, but I think it should be discussed more. In the light of the literature on EoS, large stepsizes do not only lead to faster convergence rates, but also to a stronger implicit bias towards "nice" (e.g. sparse) features. Figure 1b) does not support this claim here, but 1c) still seems to suggest that larger stepsizes can get better test loss. In consequence, I would have also liked a similar comparison of the test accuracy in figure 2 for CIFAR-10.

Additionally, the setting focuses on nearly-1-homogeneous parameterization. While the near homogeneity assumption is mild and really nice, considering 1-homogeneous networks is very restrictive and simplistic in my opinion. Again, Lyu and Li provide a general result for $L$-homogeneous parametrizations. Having a $2$-homogeneous assumption (or even general one) would largely improve the current work.
Actually, combining this assumption with the derivative condition (Assumption 2.A) makes the considered parametrization nearly linear. As a consequence, I fear that the proof of the second main result, which depends on these assumptions, does not significantly differ from the linear regression case, from a high level, abstract point of view.

**Questions:**

- Is there a specific mathematical challenge (eg wrt Lyu and Li) to prove a KKT point convergence type of result?
- Same question for extending $1$-homogeneity to $L$-homogeneity for any positive integer $L$
- Can we say more about the normalized margin in the restricted setting of Theorem 4.1 ?

---

> ### Author Rebuttal · Authors · 2024-08-06
>
> Thank you for your feedback. We address your comments below.
>
> ---
>
> **Q1**. Is there a specific mathematical challenge (eg wrt Lyu and Li) to prove a KKT point convergence type of result?
>
> **A1**. Good question. The key difficulty is that KKT points with respect to an optimization problem are not even well defined for non-homogenous predictors, so extending the KKT point convergence analysis Lyu and Li is challenging.
>
>
> Since our nearly homogeneous function $f(w;x)$ is asymptotically homogeneous, we conjecture that, under suitable conditions, GD may converge to KKT points given by a proxy homogeneous function
> $$
> \tilde f(w;x) := \left [ \lim_{t \to \infty} \frac{f(t w;x)}{t} \right] \cdot ||w||.
> $$
> There are still two obstacles to proving the above conjecture.
>
> 1. Verifying the (sub)differentiability of $\tilde f(w;x)$ is nontrivial.
> 2. The GD dynamic for learning $f$ is different to that for learning $\tilde f$. Specifically, there are examples such that $|f(w)-\tilde f(w)|$ is uniformly lower bounded by a fixed constant. Then the GD dynamic for $f$ is away from that for $\tilde f$, and might not satisfy the corresponding $(\epsilon, \delta)$ KKT conditions in [Lyu and Li].
>
> We believe additional regularity assumptions about $f$ are needed to show KKT convergence. Nonetheless, we think this is a great question, and we will discuss it in the revision.
>
> ---
>
> **Q2**. Is there a specific mathematical challenge (eg wrt Lyu and Li) to extending 1-homogeneity to
> L-homogeneity for any positive integer?
>
> **A2**. As mentioned in Line 300, this is an avenue left for future exploration. Extending our results from near 1-homogeneity to near L-homogeneity will require non-trivial modifications to both the margin functions and the stable phase conditions outlined in Theorem 2.2. Furthermore, verifying that the proposed stable phase conditions can be satisfied (even in special cases) could be challenging. We will discuss this in the revision. We think our margin improvement results in this work, which extends prior results from 1-homogeneity to near 1-homogeneity, is already a significant step toward addressing the general case.
>
> ---
>
> **Q3**. Can we say more about the normalized margin in the restricted setting of Theorem 4.1?
>
> **A3**. Good question. Although we consider linearly separable data, the predictor is nonlinear and, therefore, can achieve a normalized margin larger than the maximum $\ell_2$-margin. Additionally, we observe that in practice, while the normalized margin tends to be positive and increasing, the normalized margins for individual neurons can stay negative (please check Figure 2 in the pdf). A full characterization of the normalized margin is technically challenging even for linearly separable data. We will comment on this as a future direction in the revision.
>
> ---
>
> **Q4**. The provided results and settings might be a bit too weak in my opinion… the only implicit bias result is claiming the normalized margin is increasing in the late stable phase…. In the light of the literature on EoS, large stepsizes do not only lead to faster convergence rates, but also to a stronger implicit bias towards "nice" (e.g. sparse) features. Figure 1b) does not support this claim here, but 1c) still seems to suggest that larger stepsizes can get better test loss. I would have also liked a similar comparison of the test accuracy in figure 2 for CIFAR-10.
>
> **A4**. We respectfully disagree that our results and settings are weak. In fact, extending the implicit bias results for homogeneous predictors to general non-homogeneous predictors is an open problem listed in [Ji and Telgarsky, 2020]. Our work, by establishing the margin improvement results to general near 1-homogeneous predictors, partially solves this open problem. Furthermore, we also extend prior neural network analysis for GD with small or infinitesimal stepsizes to an arbitrary large stepsize, which is more relevant to practice. We believe the contributions of this work are already significant.
>
> While understanding the generalization benefits of large stepsizes is very interesting, this question is beyond the scope of the current paper, which focuses on margin improvement and fast optimization.
>
> Please see the attached pdf for  Figure 1 reporting the test accuracy for CIFAR-10. We will include it in the revision.
>
> ---
>
> **Q5**. ….Actually, combining this assumption with the derivative condition (Assumption 2.A) makes the considered parametrization nearly linear….
>
> **A5**. We would like to point out that Assumption 2.A is a sufficient technical condition to enable the EoS phase analysis. However, we only need Assumption 1.A for Theorem 2.2, which does not assume a lower bound on the derivative and allows the predictor to be highly non-linear.
>
> ---

---

> > ### Comment · Reviewer_RDnn · 2024-08-09
> >
> > I thank the authors for their answer. I now understand more clearly how these possible extensions still require extensive work. I think the additional discussions in the revised version will improve the quality of the paper.
> >
> > I raise my score in consequence

---

### Official Review · Reviewer_dNN4 · 2024-07-12

**Soundness:** 3
**Presentation:** 3
**Contribution:** 4
**Rating:** 7
**Confidence:** 4

**Summary:**

This work studies the phase transition (from EoS phase to stable phase) of GD with large step sizes for training two-layer networks under logistic loss. Specifically, the authors proved the following:

- If the empirical risk is below a threshold depending on the step size, GD enters a stable phase where the loss monotonically decreases and the normalized margin nearly monotonically increases.
- For linearly separable datasets, GD with an arbitrarily large step size exits the EoS phase due to the convergence of the average loss across iterations. Moreover, a tighter bound on the phase transition time is also provided.
- For linearly separable datasets, GD with an appropriately chosen step size achieves an accelerated convergence rate of $O(1/T^2)$.

**Strengths:**

This work makes significant contributions by extending existing results. The authors proved convergence with an accelerated convergence rate in the stable phase, whereas [Wu et al. (2024)] treated the linear predictor. Additionally, the authors proved margin improvement with non-homogeneous activation functions, while previous work focused on small step sizes and homogeneous activation.

**Weaknesses:**

- Despite studying two-layer neural networks, the main theorems require linear separability of datasets, except for Theorem 2.2.
- The paper missed some relevant works. For instance, [D. Barrett and B. Dherin] showed that gradient descent (in discrete time) optimizes the loss plus gradient norm and studied the modified ODE capturing this property. A stochastic variant of this dynamics was also studied by [Q. Li, C. Tai, and W. E]. Additionally, [M. Andriushchenko, A. Varre, L. Pillaud-Vivien, and N. Flammarion] and [Y. Ren, C. Ma, and L. Ying] studied the benefits of large learning rates as well. Regarding optimization in the mean-field regime, [F. Chen, Z. Ren, and S. Wang] and [T. Suzuki, D. Wu, and A. Nitanda] proved the convergence of mean-field Langevin dynamics in the finite-neuron setting.

[D. Barrett and B. Dherin] IMPLICIT GRADIENT REGULARIZATION. ICLR, 2021

[Q. Li, C. Tai, and W. E] Stochastic Modified Equations and Dynamics of Stochastic Gradient Algorithms I: Mathematical Foundations. JMLR, 2019.

[M. Andriushchenko, A. Varre, L. Pillaud-Vivien, and N. Flammarion] SGD with Large Step Sizes Learns Sparse Features. ICML, 2023.

[Y. Ren, C. Ma, and L. Ying] Understanding the Generalization Benefits of Late Learning Rate Decay. AISTATS, 2024.

[F. Chen, Z. Ren, and S. Wang] Uniform-in-time propagation of chaos for mean field Langevin dynamic. 2022.

[T. Suzuki, D. Wu, and A. Nitanda] Convergence of mean-field Langevin dynamics: time-space discretization, stochastic gradient, and variance reduction. NeurIPS, 2023.

**Questions:**

- Equation (5) in Theorem 2.2 seems to implicitly make assumptions about the data structure/distribution and the number of neurons. Can you provide any non-trivial examples other than linearly separable data? I’m wondering if this theory covers XOR or k-parity datasets, as these could be benchmarks to see the separation from the kernel regime. For instance, see the following papers:

[M. Telgarsky] Feature selection and low test error in shallow low-rotation ReLU networks. ICLR, 2023.

[T. Suzuki, D. Wu, K. Oko, and A. Nitanda] Feature learning via mean-field Langevin dynamics: classifying sparse parities and beyond. NeurIPS, 2023.
- In Figure 1(c), GD with a small step size of 0.02 seems to achieve the best test accuracy at the beginning phase. What happened?
- Can Assumption 1-C be relaxed to $\kappa \leq 1$?

**Limitations:**

The limitations have been well addressed.

---

> ### Author Rebuttal · Authors · 2024-08-06
>
> Thank you for your support.
>
> ---
>
> **Q1**. Despite studying two-layer neural networks, the main theorems require linear separability of datasets, except for Theorem 2.2.
>
> **A1**. We acknowledge that linear separability is a strong assumption. We use this mainly as a sufficient condition for two-layer neural networks, regardless of width, to reach the initial bound of the stable phase when employing large stepsizes. Additionally, our stable phase results do not need this assumption. We believe the linear separability condition can be relaxed, which is left for future work. We will comment on this more in the revision.
>
> ---
>
> **Q2**. The paper missed some relevant works. For instance, [D. Barrett and B. Dherin] showed that gradient descent (in discrete time) optimizes the loss plus gradient norm and studied the modified ODE capturing this property. A stochastic variant of this dynamics was also studied by [Q. Li, C. Tai, and W. E]. Additionally, [M. Andriushchenko, A. Varre, L. Pillaud-Vivien, and N. Flammarion] and [Y. Ren, C. Ma, and L. Ying] studied the benefits of large learning rates as well. Regarding optimization in the mean-field regime, [F. Chen, Z. Ren, and S. Wang] and [T. Suzuki, D. Wu, and A. Nitanda] proved the convergence of mean-field Langevin dynamics in the finite-neuron setting.
>
> **A2**. Thank you for bringing our attention to these works. We will cite and discuss them in detail. Specifically, the stepsize considered by [D. Barrett and B. Dherin] and [Q. Li, C. Tai, and W. E] are small or even infinitesimal, whereas our stepsize is large and causes EoS. The work by [Y. Ren, C. Ma, and L. Ying] studied special linear networks under MSE loss, while we focus on two-layer networks under logistic loss. The work by [M. Andriushchenko, A. Varre, L. Pillaud-Vivien, and N. Flammarion] studied the effect of large stepsize through experiments while we take theoretical approaches. The works by [F. Chen, Z. Ren, and S. Wang] and [T. Suzuki, D. Wu, and A. Nitanda] demonstrated the convergence of mean-field Langevin dynamics for two-layer networks with a finite but large number of neurons. In comparison, although we adopt the parameter scaling from mean-field theory, we use a different proof technique that allows any number of neurons.
>
> ---
>
> **Q3**. Equation (5) in Theorem 2.2 seems to implicitly make assumptions about the data structure/distribution and the number of neurons. Can you provide any non-trivial examples other than linearly separable data? I’m wondering if this theory covers XOR or k-parity datasets, as these could be benchmarks to see the separation from the kernel regime.
>
> **A3**. This is a good question. In fact, we only need to ensure that the dataset is realizable, that is, there exists a parameter direction $W$ such that $\inf y_i f(a W; x_i) \to \infty$ as $a\to \infty$. This indicates that the loss can be made arbitrarily small. Since XOR or k-parity datasets can be realized by two-layer networks (with Softplus, for instance), Equation (5) can be satisfied. We will discuss this in the revision.
>
> ---
>
> **Q4**. In Figure 1(c), GD with a small stepsize of 0.02 seems to achieve the best test accuracy at the beginning phase. What happened?
>
> **A4**. You are correct that GD with a small stepsize achieves the best accuracy early (after about 20000 iterations). We think this is because of a mild overfitting since we only use a small dataset of 1,979 samples, while the MLP has about 30,000 parameters.
>
> ---
>
> **Q5**. Can assumption 1-C be relaxed to $\kappa\le 1$?
>
> **A5**. In Assumption 1-C, $\kappa$ can be any fixed nonnegative number.
>
> ---

---

> > ### Comment · Reviewer_dNN4 · 2024-08-12
> >
> > Thank you for the response. I will maintain my positive evaluation.

---

### Author Rebuttal · Authors · 2024-08-06

Thank you for all your feedbacks. We add two figures in the pdf.

- Figure 1 shows the test accuracy of two-layer networks for CIFAR-10 under the same setting of Figure 2(d)-(f). The
results support the intuition that large stepsizes lead to stronger implicit biases with “nicer“ features.
- Figure 2 shows training loss and margins of a two-layer network with leaky softplus activations
on a synthetic linear separable dataset. There are  five samples in the dataset, which are $$((0.05, 1,2),1), ((0.05, -2,1),1), ((-1,0,2),-1), ((0.05,-2,-2),1),((0.05, 1,-2),1).$$ The max-margin direction is $(1,0,0)$ with a normalized margin of $0.05$.  The network only has two neurons with fixed weights $1/2$ and $-1/2$. The leaky softplus activation is $\tilde \phi(x) = ( x +  \phi(x))/2$, where $\phi$ is the softplus activation. The stepsize is $3$.  We can observe that both neurons have negative normalized margins during the training, while the network's normalized margin increases and becomes positive.

---

### Decision · Program_Chairs · 2024-09-25

**Decision:**

Accept (poster)

**Comment:**

The paper received positive reviews, with a consensus among reviewers that this is a solid extension on recent work on neural network GD dynamics. Consequently, I recommend acceptance of this paper. When preparing the camera-ready revision, please pay close attention to the reviewers’ feedback and include the additional explanations provided during the rebuttal period.